# TRIM52 maintains cellular fitness and is under tight proteolytic control by multiple giant E3 ligases

Alexandra Shulkina [1,2,3], Kathrin Hacker[1,2], Julian F. Ehrmann [3,4,7,8], Valentina Budroni [1,2,3], Ariane Mandlbauer [5], Johannes Bock[1,2,9], Daniel B. Grabarczyk [4], Genevieve Edobor[1,2], Luisa Cochella [5], Tim Clausen [4,6] & Gijs A. Versteeg [1,2] ✉

Tripartite motif 52 (TRIM52) exhibits strong positive selection in humans, yet is lost in many other mammals. In contrast to what one would expect for such a non-conserved factor, *TRIM52* loss compromises cell fitness. We set out to determine the cellular function of TRIM52. Genetic and proteomic analyses revealed TRIM52 physically and functionally interacts with the DNA repair machinery. Our data suggest that TRIM52 limits topoisomerase 2 adducts, thereby preventing cell-cycle arrest. Consistent with a fitness-promoting function, TRIM52 is upregulated in various cancers, prompting us to investigate its regulatory pathways. We found TRIM52 to be targeted for ultra-rapid proteasomal degradation by the giant E3 ubiquitin ligases BIRC6, HUWE1, and UBR4/KCMF1. BIRC6 mono-ubiquitinates TRIM52, with subsequent extension by UBR4/KCMF1. These findings suggest a role for TRIM52 in maintaining genome integrity, and regulation of its own abundance through multi-ligase degradation.

The evolutionary trajectory of organisms is often shaped by the interplay between genetic variation and environmental pressures. Understanding the function and regulation of positively-selected factors can provide insight into how different organisms adapt to their environment and thus how particular functions arose.

Evolutionary analyses have highlighted the significance of positive selection acting on genes associated with DNA damage repair, apoptosis regulation, and immune response pathways[1–4]. Particularly noteworthy among these are the tripartite motif (TRIM) protein E3 ligases, which have gained attention for their multifaceted roles in innate immunity, cellular homeostasis, and antiviral defence mechanisms[5].

Several members of the TRIM protein E3 ligase family have anti-retroviral functions and have co-evolved with the pathogens they counter[6].

Interestingly, the *TRIMS2* gene was acquired by the mammalian common ancestor, and evolved under positive selection pressure only in humans and some other primates[6]. In fact, it has been lost or pseudogenised in many other mammalian species. In contrast to what one would expect for such a non-conserved factor[6,7], *TRIMS2* ablation decreases cellular fitness and results in a p53-dependent decrease in proliferation[8,9]. Therefore, TRIM52 may play a role in limiting DNA damage arising from cell-intrinsic DNA replication stress. In line with a

---

[1]Max Perutz Labs, Vienna Biocenter Campus (VBC), Dr.-Bohr-Gasse 9, 1030 Vienna, Austria. [2]University of Vienna, Center for Molecular Biology, Department of Microbiology, Immunobiology and Genetics, Dr.-Bohr-Gasse 9, 1030 Vienna, Austria. [3]Vienna Biocenter PhD Program, a Doctoral School of the University of Vienna and the Medical University of Vienna, 1030 Vienna, Austria. [4]Research Institute of Molecular Pathology (IMP), Vienna BioCenter (VBC), Vienna, Austria. [5]Department of Molecular Biology and Genetics, Johns Hopkins School of Medicine, Baltimore, MD 21205, USA. [6]Medical University of Vienna, Vienna BioCenter (VBC), Vienna, Austria. [7]Present address: Department of Biological Chemistry and Molecular Pharmacology, Harvard Medical School, Boston, MA 02115, USA. [8]Present address: Program in Cellular and Molecular Medicine, Boston Children's Hospital, Boston, MA 02115, USA. [9]Deceased: Johannes Bock. ✉e-mail: gijs.versteeg@univie.ac.at

cell fitness-promoting function, TRIM52 expression is upregulated in several cancers[10,11].

This underpins the importance of understanding how intracellular abundance of TRIM52 is regulated, and what importance its regulation may have for its cellular function. In this context, we previously reported that TRIM52 is rapidly turned-over by the proteasome, with a half-life of just 3-4 minutes[9], positioning it as one of the most unstable proteins in the human proteome[12–16].

TRIM52 domain architecture differs from other TRIM-family members as it is comprised only of Really Interesting New Gene (RING) and B-Box domains, yet lacks a canonical Coiled-Coil domain[6]. Moreover, TRIM52 has a unique, extended RING domain. The human genome encodes approximately 600 proteins with RING domains, a key feature of most ubiquitin E3 ligase enzymes[17]. It has remained unclear whether the unusual TRIM52 RING domain has E3 activity. However, mutagenesis experiments have shown that potential TRIM52 E3 ligase activity is dispensable for its own turn-over[9], indicating that an unknown cellular machinery recognizes TRIM52 as a substrate, and marks it for degradation. This has positioned TRIM52 as an excellent model substrate to study the cellular and biochemical mechanisms by which cells mediate such rapid protein turnover.

Here we show that TRIM52 may play a role in limiting accumulation of topoisomerase 2 lesions stemming from cell-intrinsic processes such as transcription and genome replication. In the absence of TRIM52, covalent topoisomerase 2 retention on DNA is increased, ultimately culminating in downstream cell cycle arrest. Given that TRIM52 is upregulated in several cancers[10,11], we performed genetic and proteomic screens to identify its degradation machinery. This identified the giant E3 ligases BIRC6, HUWE1, and UBR4/KCMF1. Cell-based assays showed that these ligases mark TRIM52 for degradation through the highly acidic loop region in its RING domain. Mechanistically, in vitro experiments showed that TRIM52 degradation requires substrate recognition and 'seeding' of mono-ubiquitin by BIRC6, complemented by subsequent poly-ubiquitin chain extension by UBR4 and its co-factor KCMF1.

## Results

### TRIM52 is required to maintain genomic DNA integrity

TRIM52 ablation strongly diminishes cellular fitness in cell competition assays in various cell types[8,9,18], mediated by aberrant p53 activation, and subsequent cell cycle arrest[8]. Based on these findings, we hypothesized that TRIM52 may play a role in maintaining genomic DNA integrity, and that in its absence p53 is activated, resulting in cell-cycle arrest.

To gain unbiased insight into the cellular function of TRIM52 underlying this phenotype, we established a genetic screen for modifiers of TRIM52 loss. We took advantage of a human RKO cell line (p53 wildtype colon carcinoma) harbouring an inducible Cas9 cassette that has been effectively used in various genome-wide CRISPR screens (Fig. 1a)[19,20]. When we introduced sgRNAs targeting TRIM52 and induced Cas9 expression, we observed practically complete depletion of TRIM52 protein (Fig. 1b) and markedly diminished cellular fitness in a competitive proliferation assay (Fig. 1c).

We performed the genome-wide modifier screen in RKO cells harbouring a Dox-inducible Cas9-P2A-EBFP expression cassette and a genome-wide lentiviral sgRNA library, targeting one gene per cell. Into this complex cell population, we transduced a lentivirus expressing two validated sgRNAs targeting TRIM52, or two sgRNAs targeting the safe harbour locus AAVS1, as a control (Fig. 1d, Supplementary Fig. 1a). This enabled the identification of synthetically lethal genes whose mutation enhanced the TRIM52-loss-associated fitness defects. To identify TRIM52-specific modifiers, we performed a parallel counter-screen in cells with a knockout of DGCR8, a key component of the microRNA processing machinery, as its ablation reduces cell fitness to

a similar extent as ablation of TRIM52, but it is expected to act via unrelated mechanisms.

The screen did not identify any specific factors that increased cell fitness upon ablation (Supplementary data 1). However, it did identify genes whose ablation was specifically synthetically lethal with TRIM52 loss, but not with DGCR8 knock-out (Fig. 1e, f). Several high-confidence genetic interactors were associated with the error-prone, non-homologous end-joining (NHEJ) DNA damage response (Fig. 1e, f, Supplementary data 1; NHEJ1, XRCC4, and RNASEH1), indicating that this DNA repair pathway can in part compensate for functional TRIM52 loss. In line with these findings, depletion of NHEJ1 (a central factor in this DNA repair pathway) by itself did not decrease cell fitness in a competition assay, while targeting NHEJ1 in the context of TRIM52 loss diminished cell fitness more than depletion of TRIM52 alone (Fig. 1g). Depletion of other core NHEJ components Ku70 and Ku80 demonstrated a similar result (Supplementary Fig. 1b), indicating that the role of TRIM52 can in part be functionally compensated by NHEJ DNA repair. Based on this result, we concluded that TRIM52 could act upstream of multiple functionally redundant repair pathways, such as NHEJ and Homology-Dependent Repair (HDR). Alternatively, TRIM52 could act in a functionally redundant repair pathway parallel to NHEJ, such as HDR. In both cases simultaneous loss of HDR and TRIM52 would be predicted to have a non-epistatic relationship.

Therefore, we next tested in an epistasis experiment whether TRIM52 function could also be compensated for by HDR. To this end, key components of this pathway -RAD51 and BARD1- were targeted. Ablation of HDR factors in combination with TRIM52 KO had a similar additive loss-of-fitness effect as with NHEJ (Supplementary Fig. 1b), indicating a non-epistatic relationship, and that either one of these double strand break (DSB) repair pathways can function with TRIM52 in a redundant manner. This suggests that TRIM52 may play a role in maintaining genomic DNA integrity upstream of NHEJ and HDR repair pathways. This begins to explain our previous observation that loss of TRIM52 elicits p53 activation, which can be triggered by cell-intrinsic DNA damage[21–23].

### TRIM52 is required for limiting accumulation of TOP2-mediated DNA lesions

To gain further insight into which aspect of maintaining genome integrity TRIM52 is involved in, we set out to identify physical interactors of TRIM52 using TurboID proximity labeling[24] (Supplementary Fig. 2a). We generated RKO cells with a cassette for Dox-inducible expression of a TurboID-TRIM52 fusion, or TurboID-EGFP as a control[24]. After careful titration of Dox to achieve comparable protein levels (Supplementary Fig. 2b), we treated these cells with biotin for 15 min to label proteins in proximity. We performed labelling in the presence of proteasome inhibitor to stabilize the pool of likely DNA-associated TRIM52, which is otherwise low due to rapid proteasomal turn-over (Supplementary Fig. 2c). Under these conditions, TRIM52 localized to nuclear puncta (Supplementary Fig. 2d-g), reminiscent of previously reported DNA repair complexes[25]. We then isolated biotinylated proteins under denaturing conditions and used nano liquid chromatography coupled mass-spectrometry (nLC-MS/MS) for identification.

We observed a strong enrichment in proteins related to the DNA damage response and transcription regulation (Fig. 2a, b, Supplementary Fig. 2h, Supplementary data 2). Transcription can lead to DNA damage through various mechanisms, such as the formation of R-loops, and collisions between transcription and DNA replication machineries[26,27]. Pathway enrichment analysis of the proteins involved in the DNA damage response revealed a strong enrichment for factors involved HDR (Supplementary Fig. 2i, j (inner circle), and Supplementary data 3, adj. p < 0.0001). In combination with our suppressor screen results (Fig. 1e, f) this suggested that TRIM52 could play a functional role in DNA-repair or an upstream process coupled to it.

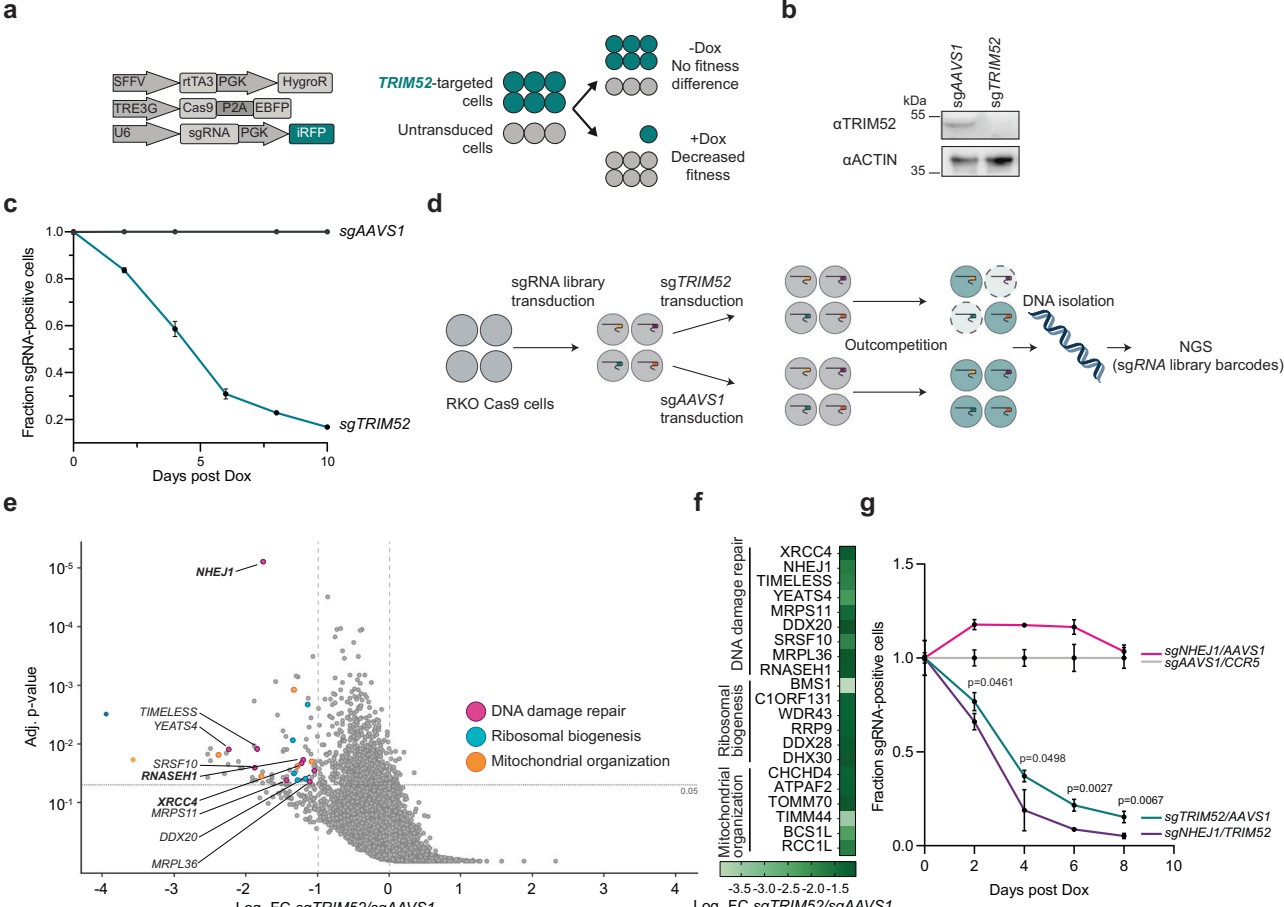

**Fig. 1 | TRIM52 is required for cell fitness. a** Schematic representation of the lentiviral expression vectors used to modify RKO Cas9 cells for knock-out and cell competition assays. **b** RKO cells harbouring Dox-inducible Cas9 were transduced with sgRNA vectors targeting *TRIM52* or the safe-harbour locus *AAVS1*. Cas9 was induced by Dox treatment for 5 days. Whole cell lysates (WCE) were analysed by WB. **c** Relative cell fitness of sg*TRIM52*-transduced cells compared to untransduced cells was determined by measuring the percentage of iRFP fluorescent cells over the indicated period and normalized to the fraction of sg*AAVS1*-transduced cells relative to untransduced cells. Data represent biological replicates as mean values +/- SD, *n* = 3. **d** Schematic representation of genetic modifier screen, grey and blue circles represent individual cells transduced with different sgRNAs. **e** RKO cells expressing Cas9 and transduced with a lentiviral genome-wide sgRNA library were further transduced with individual sgRNAs targeting *TRIM52* or *AAVS1*. Following Cas9 induction, cells were grown for 12 doublings and integrated sgRNA-coding sequences were analysed by NGS analysis of gDNA, sgRNA enrichment calculated by

MAGeCK analysis, and log2-fold change and adjusted p-value plotted relative to library representation determined in the unselected cell population. Factors on the left-hand side of the plot are identified as genes which lead to synthetic lethality upon ablation in *TRIMS2*-targeted cells. Non-*TRIM52*-specific genes were filtered out by comparison with data from *DGCR8*-targeted cells. Factors involved in DNA-damage repair are marked in pink and labelled by name. **f** Filtered *TRIM52* ablation-specific genes were selected and their Log2 fold change compared to *AAVS1*, plotted in a heatmap, and grouped by their function. **g** Relative cell fitness of sg*TRIM52*, sg*NHEJ1* and sg*TRIM52*/sg*NHEJ1* transduced cells compared to untransduced cells was determined by measuring the percentage of iRFP fluorescent cells over the indicated period. sg*TRIM52* fractions were normalized to sg*AAVS1*/*CCR5*-transduced cells, relative to untransduced cells. Data represent biological replicates as mean values +/- SD, *n* = 3. Data analysed by unpaired two-sided t-test comparing means of TRIM52/AAVS1 and NHEJ1/TRIM52, with FDR adjustment using Benjamini, Krieger, and Yekutieli. Source data are provided as a Source Data file.

Two of the strongest identified TRIM52 interactors (Fig. 2a, b), Tyrosyl-DNA Phosphodiesterase 2 (TDP2) and Zinc Finger Protein 451 (ZNF451), are not part of the core DNA-repair machinery but rather play key roles upstream of repair pathways in specifically sensing and resolving topoisomerase 2 (TOP2) lesions[28–32]. TOP2 resolves topological problems in DNA that originate from DNA replication, transcription, and chromosome dynamics[33]. TOP2 introduces DSBs to reduce torsional stress, and becomes covalently linked to the 5′ phosphate via a tyrosine residue[28–30]. For error-free repair of these DSBs, TOP2 must be removed from DNA, which is achieved through the action of TDP2 and the SUMO E3 ligase ZNF451 that facilitates TDP2 hydrolase activity on stalled TOP2 cleavage complexes[28–32]. This leaves 5′ phosphates free for repair by HDR and NHEJ (Fig. 2c)[34–36]. These activities are critical for cell fitness, as their inhibition results in p53 activation and cell cycle arrest, even in the absence of exogenous DNA damage[28,29,31,37], as seen upon loss of *TRIM52*[8]. Therefore, we

hypothesized that TRIM52 plays a role in the repair of TOP2-associated DSBs, serving an important function in the TDP2-dependent TOP2 cleavage complex removal pathway (Fig. 2c).

To test this hypothesis, we measured TOP2-DNA covalent adducts using the 'Rapid Approach to DNA Adduct Recovery' (RADAR) assay[38,39]. In this assay, cellular DNA with covalently associated protein complexes is precipitated, after which TOP2 levels are measured by slot blot. We reasoned that if TRIM52 plays a functionally important role in limiting the accumulation of TOP2 lesions, its ablation would sensitize cells to TOP2 poisons, such as Etoposide (ETO). We exposed cells in which either the safe-harbour loci *AAVS1*/*CCR5* or *TRIMS2* were targeted with sgRNAs to sub-saturating concentrations of ETO, then measured TOP2 cleavage complexes (TOP2cc) associated with DNA by RADAR assay. Under untreated conditions, *TRIM52*-targeting showed a trend for increased DNA-associated TOP2cc (Fig. 2d). ETO treatment increased the levels of TOP2cc by 5.5-fold in sg*AAVS1*/*CCR5* control

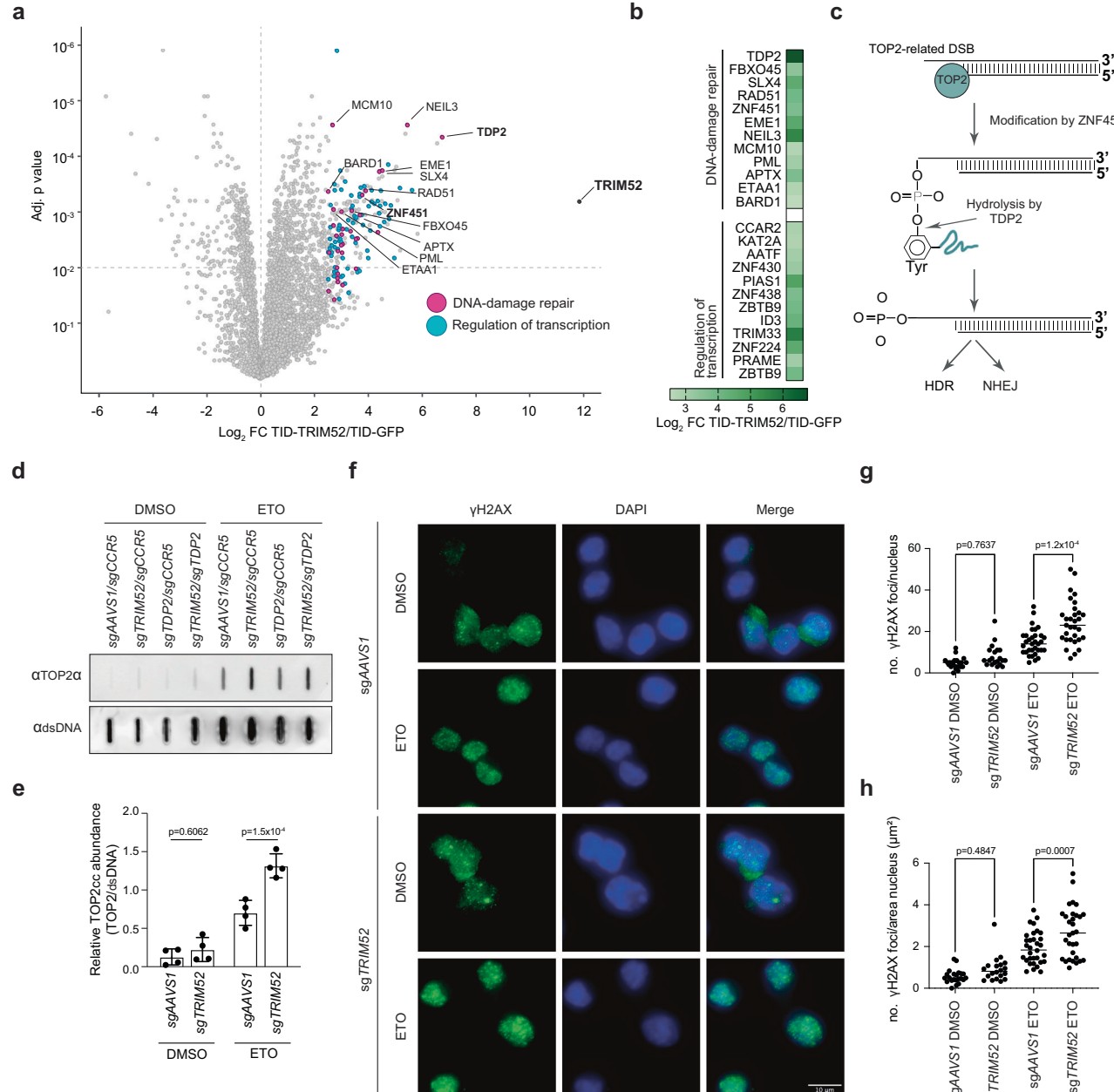

**Fig. 2 | TRIM52 interacts with the DNA repair machinery. a** A TurboID-TRIM52 fusion was expressed, cells treated with proteasome inhibitor for 5 h., and biotin for the last 15 min. Biotinylated proteins were purified under denaturing conditions and analysed by mass-spectrometry. Factors involved in DNA-damage repair are marked in pink and labelled by name. Data represent biological replicates, *n* = 3. **b** Putative interactors of TRIM52 with *p* value < 0.05 and Log2 fold change > 2.5 were selected, analysed by gene ontology enrichment analysis, and top hits per ontology category plotted in a heatmap by their Log2 fold change enrichment compared to the TurboID-EGFP control. Statistical analysis was conducted using moderated t-statistics via the limma-trend method in R[95] and applying the Benjamini–Hochberg multiple testing correction. Data represent biological replicates as mean values +/- SD, *n* = 3. **c** Schematic representation of TOP2 cleavage complex resolution. **d** RKO-Cas9 cells transduced with sgRNAs targeting safe

harbour loci *AAVS1/CCR5, TRIM52, TDP2* or both were treated with Dox for 5 days to induce Cas9. Cells were treated with etoposide (ETO) for 2 h. Samples were harvested and subjected to RADAR analysis. Isolated protein-DNA covalent complexes were analysed by slot blot, **e** quantified and normalized to dsDNA levels. Data represent biological replicates as mean values +/- SD, *n* = 4. Data were analysed by 2-way ANOVA. **f** RKO-Cas9 cells transduced with sgRNAs targeting the safe harbour locus *AAVS1* or *TRIM52* were treated with Dox for 5 days to induce Cas9. Cells were treated with DMSO or ETO for 2 h., fixed, and stained with anti-γH2AX antibody. The number of γH2AX foci was determined by immunofluorescence microscopy, and **g** plotted, and **h** plotted normalized to the surface area of the corresponding nuclei. Scale bar: 10 μm. The number of foci were counted for each biological sample. Data represent *n* = 3 technical replicates, analysed by 2-way ANOVA. Source data are provided as a Source Data file.

cells, which was further increased ~2-fold in *TRIM52*-targeted cells (Fig. 2e), indicating the requirement of TRIM52 for efficient hydrolysis of TOP2-DNA phospho-tyrosyl bonds.

Failure to resolve trapped TOP2ccs leads to DSB formation. Therefore, we tested whether ablation of *TRIM52* results in increased amount of DSBs in nuclei by γH2AX analysis.

*AAVS1-* or *TRIM52*-targeted cells were treated with DMSO or etoposide and stained for γH2AX (Fig. 2f). In the presence of etoposide, loss of *TRIM52* significantly increased the number of γH2AX-marked DNA lesions (Fig. 2g, h), underpinning the importance of TRIM52 for limiting the accumulation of TOP2ccs.

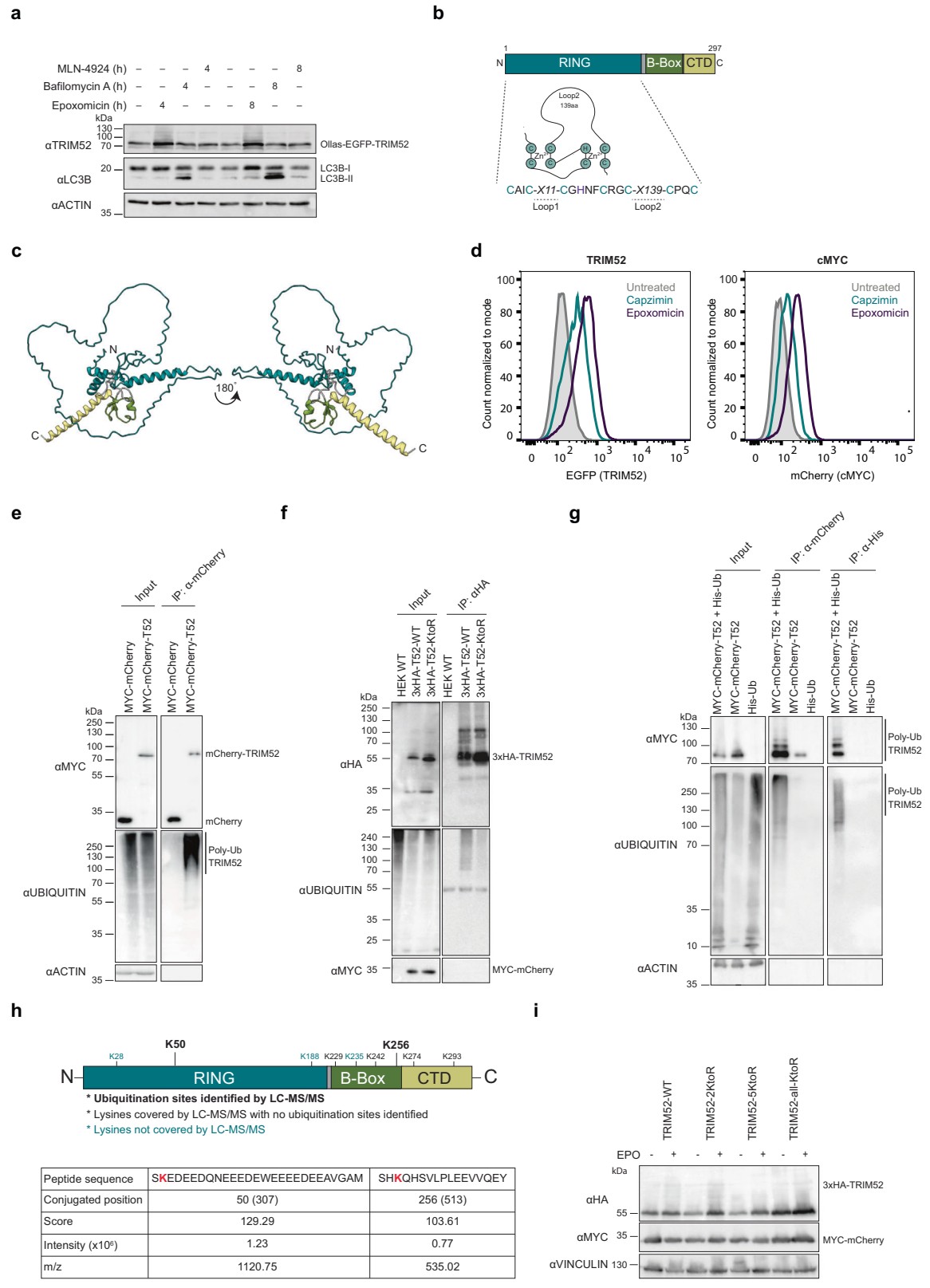

Taken together, these results show that TRIM52 plays a role in limiting the accumulation of TOP2-dependent DNA lesions. These findings help explain previous observations showing that loss of *TRIM52* results in cell-intrinsic activation of p53, and ultimately cell cycle arrest[8].

## TRIM52 is degraded by the ubiquitin-proteasome system

Considering TRIM52's function in DNA repair, and the fact that its levels are upregulated in several cancers[10,11], we next set out to understand how its abundance is regulated. First, to determine which cellular degradation pathways contribute to TRIM52 turn-over, we

**Fig. 3 | TRIM52 is degraded by the ubiquitin-proteasome system. a** RKO cells expressing Ollas-tagged EGFP-TRIM52 were treated for 4 or 8 h. with the proteasome inhibitor epoxomicin (EPO), the lysosome inhibitor bafilomycin A, or the neddylation inhibitor MLN-4924. WCEs were analysed by WB. **b** Schematic representation of TRIM52 domain organization. **c** AlphaFold2 model for TRIM52 structure prediction. **d** RKO cells expressing Ollas-tagged EGFP-TRIM52 or mCherry-cMYC fusion proteins were treated for 5 h. with the proteasomal 20S catalytic core particle inhibitor EPO, or 19S regulatory particle-specific inhibitor capzimin. Ollas-EGFP-TRIM52 and mCherry-cMYC protein levels were determined by measuring EGFP or mCherry fluorescence using flow cytometry, and their mean fluorescence intensity (MFI) plotted. **e** RKO cells expressing MYC-tagged mCherry, or MYC-tagged mCherry-TRIM52 were treated with epoxomicin for 5 h., lysed

under denaturing conditions, RFP-trap IPs performed, and analysed by WB for endogenous ubiquitin. **f** HEK-293T cells transfected with 3xHA-tagged TRIM52 or TRIM52-KtoR in which all lysines were mutated to arginines were treated with epoxomicin for 5 h., lysed under denaturing conditions, TRIM52 was immunoprecipitated, and its ubiquitination analysed by WB. **g** HEK-293T cells expressing MYC-tagged mCherry or mCherry-TRIM52, and His-tagged ubiquitin were treated with epoxomicin for 5 h., and dual-affinity purified by RFP-trap and NiNTA. Eluates were analysed by WB. **h** Ubiquitination sites identified by nLC-MS/MS plotted on schematic TRIM52 domain representation. **i** HEK-293T cells were transfected with 3xHA-tagged TRIM52 WT and TRIM52 lysine-to-arginine mutants, treated with EPO for 5 h., and lysates analysed by WB. Source data are provided as a Source Data file.

treated RKO cells expressing Ollas-tagged EGFP-TRIM52 with the proteasome inhibitor epoxomicin, the lysosome inhibitor bafilomycin A, or the neddylation inhibitor MLN-4924, which inhibits Cullin ubiquitin E3 ligases. We then analysed Ollas-EGFP-TRIM52 protein levels in whole cell extracts (WCE) by Western blot (WB), and by flow cytometry. Proteasome inhibition by epoxomicin increased Ollas-EGFP-TRIM52 protein levels by 2.5-fold in WB (Fig. 3a, and Supplementary Fig. 3a), and 3.6-5.2-fold by flow cytometry (Supplementary Fig. 3b). None of the other inhibitors affected TRIM52 concentrations, indicating that it is predominantly degraded through proteasomal degradation.

TRIM52 has a long, low complexity loop 2 within its RING domain (Fig. 3b) that is predicted to be unstructured by AlphaFold2.0 (Fig. 3c, Supplementary Fig. 3c, d). To test whether TRIM52 is degraded by 20S or 26S proteasomes, we compared the effect of the 19S regulatory particle inhibitor capzimin (which exclusively inhibits 26S proteasomes)[40] with the 20S core particle inhibitor epoxomicin (which inhibits both 20S and 26S proteasomes). Since capzimin is a less effective inhibitor than epoxomicin[40], we compared its effects on TRIM52 to cMYC, which is degraded in a ubiquitin- and 26S proteasome-dependent manner[41,42]. We treated cells expressing Ollas-tagged EGFP-TRIM52 or mCherry-cMYC fusion proteins with the indicated proteasome inhibitors, then analysed the effects on their steady-state levels by flow cytometry. Both inhibitors comparably increased TRIM52 and cMYC levels (Fig. 3d and Supplementary Fig. 3e). Epoxomicin increased TRIM52 levels by 1.7-fold and cMYC by 2.7-fold. As expected, the effect of capzimin was less pronounced (Fig. 3d, and Supplementary Fig. 3e): 0.5-fold for TRIM52 and 0.75-fold for cMYC. Since the relative effects of both inhibitors were comparable between both substrates, we concluded that, like cMYC, TRIM52 is predominantly degraded in a 26S proteasome-dependent manner.

To test whether TRIM52 is ubiquitinated in cells, we immunoprecipitated MYC-tagged mCherry-TRIM52 from RKO cell lysates and checked for associated poly-ubiquitin chains by WB. Consistent with 26S proteasomal degradation (Fig. 3d, and Supplementary Fig. 3e), MYC-mCherry-TRIM52 was strongly ubiquitinated, whereas a MYC-tagged mCherry control was not detectably modified (Fig. 3e). Lysine is the most prominent residue for post-translational modification by ubiquitin. Therefore, we compared the ubiquitination of HA-tagged WT TRIM52 with a mutant in which all lysines were mutated to arginines. Consistent with reduced degradation, steady-state levels of the lysine-less TRIM52 mutant were increased relative to its WT counterpart, whereas its ubiquitination was reduced by 70% (Fig. 3f, Supplementary Fig. 3f). It is currently unclear why lysine-less TRIM52 was not fully stabilized; it is possible that multimerization with endogenous TRIM52, or non-lysine ubiquitination contribute to its turn-over.

To identify the sites of ubiquitination in TRIM52, we tandem-purified MYC-tagged mCherry-TRIM52 from cells expressing his-tagged ubiquitin (Fig. 3g) and performed nLC-MS/MS identification of peptides with di-Gly ubiquitin remnants[43,44]. This revealed a strong presence of di-Gly residues on ubiquitin itself on K11, K29, K33, K48, and K63, with K48 being >10 fold as abundant as any of the other

detected linkages (Supplementary data 4), indicating K48-linked chains to be the major poly-ubiquitin modification on TRIM52. In addition, we identified two high-confidence di-Gly sites on TRIM52: one site within its loop 2 region of the RING domain (K50) and another in the BBox domain (K256) (Fig. 3h; Supplementary data 4). We detected peptides covering four of the remaining lysines (K229, K242, K276, K293) by nLC-MS/MS, but they were not significantly ubiquitinated. No peptides containing K28, K188, and K235 were identified by nLC-MS/MS, for which it thus remained unclear whether they were ubiquitinated (Fig. 3h) and could contribute to degradation.

To test the functional importance of these lysine residues for TRIM52 degradation, we tested sensitivity to proteasome inhibition for TRIM52 mutants in which groups of lysines were mutated to arginines. Mutation of the two identified lysines (2KtoR; K50/K256) did not result in TRIM52 stabilization, nor did additional mutation of the three lysines that were not covered by nLC-MS/MS analysis (5KtoR; Fig. 3i). Only mutation of all lysines (all-KtoR) increased steady-state TRIM52 levels, and rendered it insensitive to epoxomicin treatment (Fig. 3i).

Together, these results show that TRIM52 is K48 poly-ubiquitinated at a minimum of two predominant lysine residues. Removal of these sites was not sufficient to prevent TRIM52 degradation, suggesting that the E3 ligase machinery targeting TRIM52 can modify non-dominant sites in their absence.

## TRIM52 is targeted for degradation by multiple giant E3 ligases

To identify the factors responsible for the ultra-rapid degradation of TRIM52, we performed another genetic screen in the RKO-Cas9-EBFP cell line. In this case, we introduced a stably expressed dual protein stability reporter (Fig. 4a). This bicistronic reporter was designed to translate stable MYC-tagged mCherry and unstable Ollas-tagged EGFP-TRIM52 in equimolar amounts through a P2A ribosomal skip site[45], yet accumulate low EGFP-TRIM52 steady-state levels as a result of its degradation (Fig. 4a).

We selected a monoclonal cell line that had sufficiently high EGFP-TRIM52 levels to enable screening by flow cytometry and yet mirrored endogenous TRIM52 instability. Specifically, we determined that levels of EGFP-TRIM52, but not mCherry, were increased by proteasome inhibition (Supplementary Fig. 4a, c). In addition, the turn-over of EGFP-TRIM52 measured upon translation inhibition was dramatically faster than that of mCherry (Supplementary Fig. 4d, e). Even though overexpression did extend the half-life of the reporter to 30 min, compared to 3.3 min for endogenous TRIM52, this is still a very unstable protein. For reference, endogenous cMYC is considered highly unstable with a half-life of 15–30 min[12]. Therefore, we concluded it to be a suitable reporter for screening. Given that we established a new monoclonal line, we also confirmed the efficiency and inducible control of Cas9-mediated gene editing in this cell line (Supplementary Fig. 3f).

To screen for factors that induce the degradation of TRIM52, we transduced a lentiviral sgRNA-encoding library into our reporter cell line at a low multiplicity of infection to ensure targeting of only one gene per cell (Fig. 4b, Supplementary Fig. 4g). The sgRNA library was

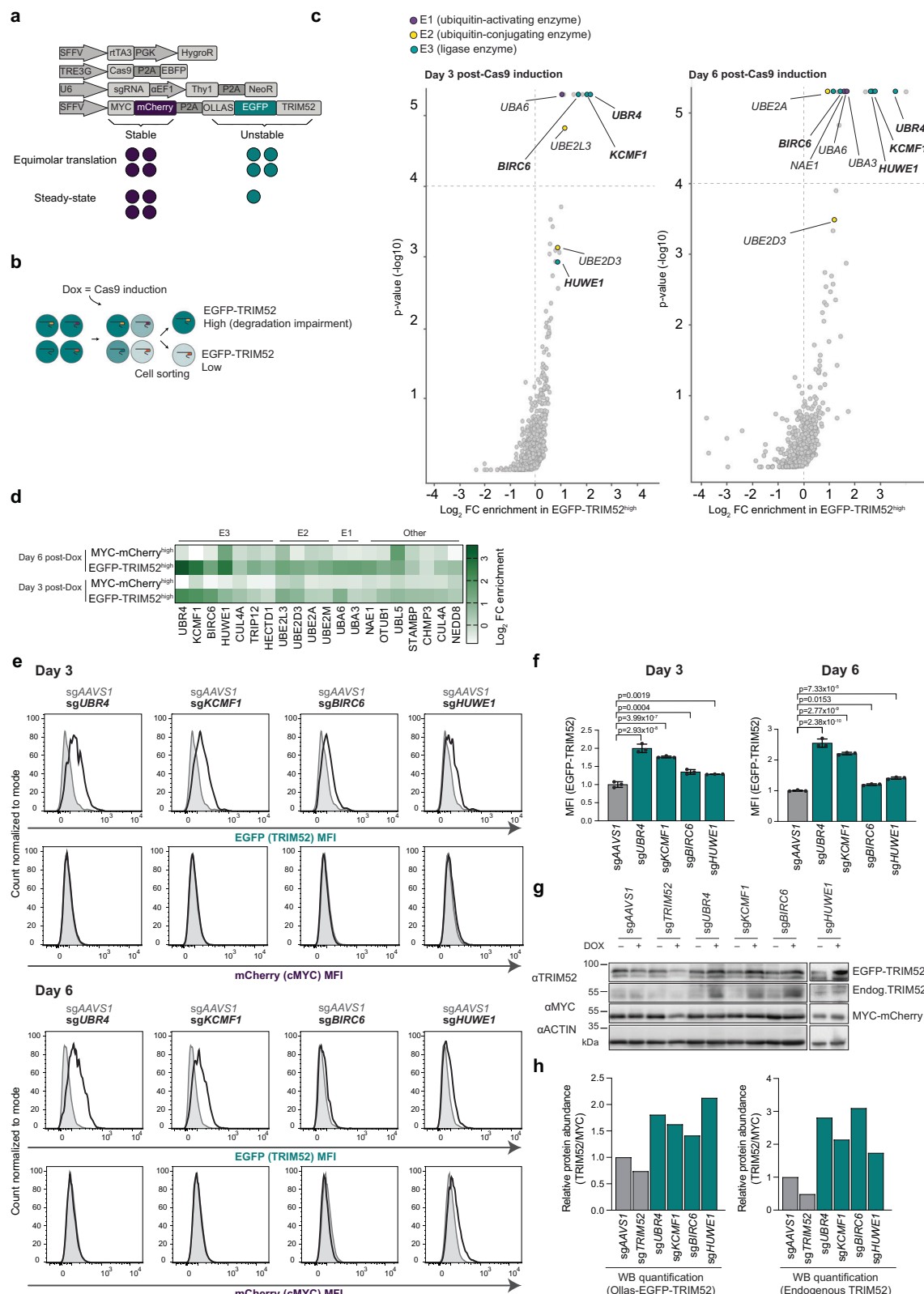

designed to target ~1000 genes, including E1 activating enzymes, E2 conjugases, E3 ligases, deubiquitinating enzymes, and ubiquitin-interacting proteins. At two time points after Cas9 induction (3 and 6 days), we used FACS to select the top 1-2% EGFP[high] cells, with potential defects in TRIM52 degradation. In parallel, we collected a comparable control cell pool for mCherry to exclude genes that generally increase protein abundance (Fig. 4b and Supplementary Fig. 5a).

The sgRNA loci present in these various cell populations were amplified, quantified by NGS, and plotted relative to their representation in unsorted cell pools (Fig. 4c, d).

This screen detected a strong enrichment of select E1, E2 and E3 ligases in the EGFP-TRIM52[high] cells that were absent in the mCherry[high] control population (Fig. 4c, d, Supplementary Fig. 5b). Specifically, we identified two E1 activating enzymes, UBA3 and UBA6, the E2

**Fig. 4 | TRIM52 is targeted for degradation by multiple giant E3 ligases.**
**a** Schematic representation of the expression vectors in the screening cell line. mCherry and EGFP-TRIM52 are expressed in an equimolar manner, yet EGFP-TRIM52 accumulates at low steady-state levels, resulting from its rapid proteasomal turn-over. **b** Schematic representation of the genetic screen to identify TRIM52 regulators. EGFP$^{high}$ cells (blue circles) with potential knock-outs in factors involved in TRIM52 degradation were collected by FACS, and their integrated sgRNA CDSs quantified by NGS, relative to those from unsorted cell pools. **c** Screening cells were transduced with an sgRNA library targeting ubiquitin-related genes and treated with Dox for 3 or 6 days. Cells expressing the highest and lowest 1·2% of each fluorophore were collected by FACS, their integrated sgRNA CDSs quantified by

NGS, and sgRNA enrichment calculated by MAGeCK analysis, and log2-fold change and adjusted p-value plotted. **d** Heatmap displaying the Log2 fold change for genes enriched in the EGFP-TRIM52$^{high}$ sorted population after exclusion of genes enriched in mCherry$^{high}$ population on days 3 and 6. **e** RKO-Cas9 cells expressing MYC-mCherry-P2A-Ollas-EGFP-TRIM52 (teal) or mCherry-cMYC-P2A-EBFP (purple) as a control were transduced with lentiviral vectors expressing the indicated sgRNAs. Cas9 expression was induced for 6 days, after which EGFP-TRIM52 and mCherry-cMYC protein levels were quantified by flow cytometry (**f**), or analysed by WB (**g–h**). **f** MFI plotted for flow cytometry samples. Data represent biological replicates as mean values +/- SD, $n = 3$. Data were analysed by 1-way ANOVA. **g** WB samples, and **h** quantification by densitometry. Source data are provided as a Source Data file.

---

conjugating enzymes UBE2D3, UBE2L3, UBE2A and UBE2M, and three high-confidence giant E3 ligases: BIRC6, HUWE1 and UBR4 with its known interactor KCMF1[46] (Fig. 4c, d, Supplementary data 5). We validated the effect of the three identified E3 ligases in independently generated polyclonal cell pools (Fig. 4e, f). Moreover, to test their relative specificity for controlling TRIM52 turn-over, we targeted the same E3 ligases in cells expressing an mCherry-tagged version of the proteasome-targeted transcription factor cMYC. In contrast to the effect on EGFP-TRIM52, knock-out of *UBR4/KCMF1* or *BIRC6* had no measurable influence on the steady-state levels of mCherry-cMYC (Fig. 4e; rows 2 and 4). This was consistent with similar genetic screens that identified degradation-regulators of cMYC or another unrelated transcription factor, IRF1[47]. HUWE1 has been previously identified as a regulator of cMYC degradation[47,48], and consistent with these findings, its knock-out did moderately increase cMYC protein levels, especially at day 6 post-Cas9 induction (Fig. 4e). However, we still included HUWE1 as a candidate TRIM52 regulator considering its broad substrate range[49,50].

To confirm that these candidate E3 ligases are bona fide regulators of endogenous TRIM52, and not only of the TRIM52 reporter used for genetic screening, we also showed that loss of the E3 ligases increased endogenous TRIM52 levels (Fig. 4g, h). To test whether the increased TRIM52 levels indeed stemmed from increased protein stability, we targeted the individual E3 ligases in RKO cells stably expressing HA-tagged TRIM52 and an internal MYC-tagged mCherry control, and determined protein levels by WB at different timepoints after protein synthesis inhibition by CHX (Supplementary Fig. 5c). The half-life of HA-TRIM52 in the sg*AAVS1* control was 8.3 min and increased 2-5-fold to 20-40 min in the different E3 ligase knock-outs (Supplementary Fig. 5d). Taken together, we identified BIRC6, HUWE1, and UBR4/KCMF1 as E3 ligases that functionally mediate TRIM52 protein turn-over.

## BIRC6, HUWE1, and UBR4/KCMF1 target the extended loop 2 region in the TRIM52 RING domain

Having identified E3 ligases that mediate TRIM52 turn-over, we next set out to identify which TRIM52 protein domains and features confer its instability, as they may provide insight into how the identified E3 ligases recognize TRIM52 as a substrate. To this end, we transfected HEK-293T cells with vectors expressing mCherry-tagged WT TRIM52, TRIM52 mutants, or individual protein domains in isolation (Fig. 5a). We gated cells for similar levels of the independently expressed EBFP internal control (Fig. 5a). As a measure of the instability of the tested proteins, we treated these cell pools with proteasome inhibitor, and then determined the levels of their mCherry-tagged fusion proteins by flow cytometry. Consistent with our previous findings that TRIM52 is rapidly degraded, mCherry-TRIM52 significantly increased upon epoxomicin treatment (Fig. 5a) compared to the mCherry control. Instability was lost upon removal of the extended loop 2 region in the TRIM52 RING domain (ΔLoop2), indicating this region to be the main determinant of TRIM52 degradation. Consistent with this notion, the RING domain in isolation was sufficient to confer instability to

mCherry, but not in the absence of loop 2 (Fig. 5a; compare samples 4 and 5 to mCherry).

To test whether the highly acidic composition of the loop 2 region is required for TRIM52 turn-over, we mutated acidic D/E residues to neutral or basic residues in a manner that is predicted to maintain the disordered nature of the loop 2 region. Conversion of the loop 2 region into a more basic and positively charged variant completely stabilized the fusion protein (Fig. 5a). Likewise, a RING mutant carrying a neutral loop 2 was more stable than the WT RING domain as evidenced by a significant reduction in its stabilization upon proteasome inhibition (Fig. 5a). We then investigated whether the acidic/negative unstructured loop 2 region of TRIM52 was sufficient to confer instability in isolation. Indeed, a fusion of just loop 2 to mCherry or EGFP rendered them sensitive to proteasome inhibition, whereas a basic variant, or an unstructured region from CY2B did not confer instability (Fig. 5a and Supplementary Fig. 6a).

Lastly, we reasoned that the D/E-rich nature of the TRIM52 loop 2 region per se could be conferring instability, independent of its encoded protein sequence. Such compositionally biased regions are present only in few select other proteins, including in the cellular protein DAXX. To functionally test whether sequence composition independent of its encoded peptide played a role in protein turn-over, the D/E-rich region of DAXX, which has a composition and length similar to TRIM52 loop 2 (Supplementary Fig. 6b), was fused to mCherry. In contrast to TRIM52 loop 2, a fusion of the acidic region of DAXX did not confer instability to mCherry (Fig. 5a).

We concluded that the loop 2 region in the TRIM52 RING is the major determinant of TRIM52 degradation, and likely a stand-alone degron. While its biased amino acid composition is required for its destabilizing effect, additional features beyond charge and disorder likely play a role, as a compositionally comparable region from DAXX did not confer instability (Fig. 5a and Supplementary Fig. 6b).

To test whether TRIM52 ubiquitination is dependent on its loop 2 region, we isolated HA-tagged TRIM52 from cells and analysed its ubiquitination by WB. Indeed, full-length HA-TRIM52 was strongly ubiquitinated, while almost all ubiquitination was lost in the absence of loop 2 (Fig. 5b). A TRIM52 variant with a neutral loop 2 was still ubiquitinated comparable to wild-type HA-TRIM52 (Fig. 5b), consistent with the finding that it was unstable (Fig. 5a, sample 6). To test whether the RING loop 2 is also the region through which the E3 ligases mediate TRIM52 degradation, we ablated the individual E3 ligases in HEK-293T cells expressing either full-length WT TRIM52, a TRIM52-ΔLoop2 mutant, only the RING domain, or the RING domain lacking the loop 2 region. Ablation of the identified E3 ligases (Supplementary Fig. 6c) significantly increased the levels of full-length TRIM52 and its RING domain in isolation by 1.6-3.1-fold (Fig. 5c, set 1 and 3, and Supplementary Fig. 6d), while neither full-length TRIM52 nor the RING domain in isolation were stabilized upon removal of the loop 2 region (Fig. 5c, set 2 and 4). Therefore, both TRIM52 ubiquitination and degradation of TRIM52 by the identified giant E3 ligases are dependent on the loop 2 region.

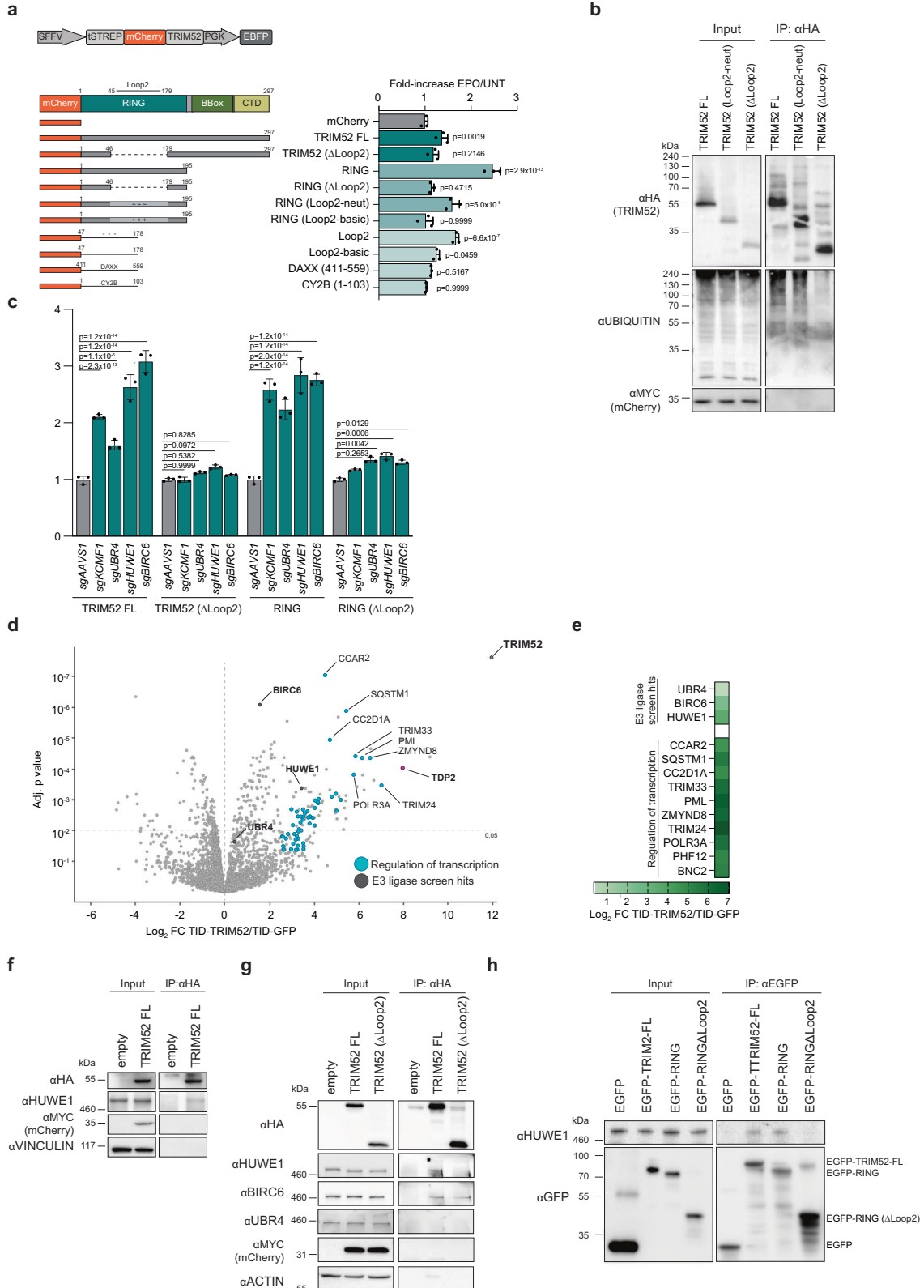

## BIRC6 and HUWE1 bind to TRIM52

We reasoned that there are three main possibilities for how the loop 2 region could contribute to TRIM52 turn-over: **i)** it is the site of E3 binding/recognition, **ii)** it is the site of ubiquitination, or **iii)** it is the site for both E3 binding and ubiquitination. To distinguish between the first two options and the last option, we utilized the finding that

fusion of loop 2 to mCherry and EGFP rendered these otherwise stable fluorophores in part unstable (Fig. 5a, Supplementary Fig. 6a). We thus predicted that if a given E3 ligase either binds or ubiquitinates outside of loop 2, an mCherry-loop2 construct would not be affected upon E3 ligase ablation (either option 1 or 2). However, if a given E3 mediates TRIM52 targeting in a loop

**Fig. 5 | BIRC6, HUWE1 and UBR4/KCMF1 target the extended loop 2 region in the TRIM52 RING domain. a** HEK-293T cells were transfected with plasmids expressing the indicated TwinStrep-mCherry-tagged TRIM52 expression constructs and an internal EBFP as a control. Cells were treated with proteasomal inhibitor epoxomicin (EPO) for 5 h., after which protein levels were quantified by flow cytometry, and normalized to EBFP levels and to corresponding untreated (UNT) samples and plotted. Data represent biological replicates as mean values +/- SD, $n = 3$. Data were analysed by 1-way ANOVA. **b** HEK-293T cells were transfected to express HA-tagged WT TRIM52, TRIM52 neutral loop2 (Loop2-neut) or TRIM52 in which the loop 2 region is reduced to the size of other RING proteins (ΔLoop2), treated with EPO for 5 h, TRIM52 immunoprecipitated, and ubiquitination levels analysed by WB. **c** HEK-293T cells constitutively expressing Cas9 and sgRNAs targeting the indicated genes were transfected with plasmids expressing TwinStrep-mCherry-tagged TRIM52, TRIM52 (ΔLoop2), RING, or RING (ΔLoop2), as well as an internal control EBFP. Protein levels were quantified by flow cytometry, normalized to EBFP and the sg*AAVS1* control, and plotted. Data represent biological replicates

as mean values +/- SD, $n = 3$. **d** RKO cells expressing a TurboID-TRIM52 fusion protein were incubated with biotin for 15 min., after which biotinylated proteins were purified under denaturing conditions, and analysed by mass-spectrometry. Statistical analysis was conducted using moderated t-statistics via the limma-trend method and applying the Benjamini–Hochberg multiple testing correction. Data represent $n = 3$ biological replicates. **e** Heatmap displaying the Log2 fold change of selected interactors of TRIM52. **f** HEK-293T cells expressing 3xHA-tagged TRIM52 were treated with EPO for 5 h., after which TRIM52 was immunoprecipitated, and analysed by WB for co-immunoprecipitation with HUWE1. **g** HEK-293T cells expressing 3xHA-tagged TRIM52 were treated with EPO for 5 h, after which TRIM52 was immunoprecipitated, and analysed by WB for co-immunoprecipitation with HUWE1, BIRC6, and UBR4. **h** Purified TwinStrep-tagged EGFP-TRIM52, EGFP-RING and EGFP-RINGΔLoop2 were incubated with purified human HUWE1 for 3 h. Twin-Strep-EGFP-TRIM52 was then immunoprecipitated from the samples using EGFP-trap beads and analysed for complex formation with HUWE1 by WB. Source data are provided as a Source Data file.

2-autonomous manner (option 3), then such a construct should be stabilized in the absence of the E3 ligase.

To test this, we ablated the E3 ligases in cells expressing either mCherry or mCherry-loop2 and determined the protein levels by flow cytometry. As expected, none of the knock-outs influenced mCherry levels (Supplementary Fig. 6e). However, *UBR4* and *HUWE1* ablation significantly increased mCherry-loop2 levels (Supplementary Fig. 6e), indicating that loop 2 alone is sufficient for targeting by these two E3s. In contrast, loss of *BIRC6* did not significantly affect mCherry-loop2 (Supplementary Fig. 6e), suggesting that BIRC6 either binds TRIM52 in loop 2 but ubiquitinates it outside of this region, or vice versa.

In addition, we also tested whether TRIM52 auto-ubiquitination plays a role in its degradation. We therefore expressed a catalytically inactive TRIM52 allosteric linchpin[51] mutant (TRIM52-R187A), and measured its turn-over after translation block. Catalytically inactive TRIM52 was rapidly turned over with a half-life of 31.6 min, comparable to its wildtype counterpart, and its protein levels increased in response to epoxomicin in a similar fashion to wild-type TRIM52 (Supplementary Fig. 6f). These results demonstrate that TRIM52's own E3 ligase activity is dispensable for its turnover, as previously reported[9].

Genetic interactors identified in our screen (Fig. 4c-d) could affect TRIM52 turn-over **i)** directly by binding it as a substrate, **ii)** acting as ubiquitin chain extending E4 enzymes on (mono-) ubiquitinated substrates, or **iii)** indirectly by affecting other proteins. We reasoned that cellular complex formation between TRIM52 and any of the identified E3 ligases would indicate that TRIM52 is recognized as a direct substrate. Lack of such cellular complex formation would be consistent with possible E4 function or indirect effects, yet would not rule out TRIM52 as a direct substrate as some interactions are transient.

To identify the E3 ligases interacting with TRIM52 in an unbiased manner, we performed additional TurboID proximity labelling experiments[24]. In contrast to the TurboID samples generated for assessing the functional role of TRIM52 (Fig. 2a-b), these samples were not generated in the presence of proteasome inhibitor, in order to maximize capture of the total steady-state interactome. Consistent with the earlier TurboID data (Fig. 2a-b, and Supplementary data 2), BIRC6 and HUWE1 were significantly enriched in TurboID-TRIM52 samples, compared to TurboID-EGFP controls (Fig. 5d-e, and Supplementary Fig. 6g). While UBR4 peptides were detected in TurboID-TRIM52 samples, they were not significantly enriched over the TurboID-EGFP control (Fig. 5d-e). KCMF1 peptides were not detected in the TurboID-TRIM52 samples at all. Consistent with a role of TRIM52 in limiting the accumulation of DNA damage lesions stemming from cell intrinsic processes (Fig. 2a-g), we also identified interactors involved in DNA-dependent transcription (Fig. 5d-e, Supplementary Fig. 6h).

Together, these findings indicate that BIRC6 and HUWE1 likely form a complex with TRIM52 in cells, making them strong candidates

for E3 ligases directly ubiquitinating TRIM52. Although the absence of detectable UBR4/KCMF1 interaction does not exclude their direct ubiquitination of TRIM52, it suggested that a complex of UBR4/KCMF1 could contribute to TRIM52 degradation in an E4 ligase capacity through ubiquitin chain extension, or another indirect manner.

Results presented in Supplementary Fig. 6e indicated that HUWE1 likely binds inside loop 2 and ubiquitinates inside of it, whereas BIRC6 either binds inside or outside loop 2. To test this by co-IP, we isolated full-length TRIM52 or its Δloop2 mutant from cells and analysed the association of HUWE1 and BIRC6. Full-length TRIM52 did in fact co-IP both HUWE1 and BIRC6 (Fig. 5f-h, Supplementary Fig. 6i). While HUWE1 binding was strongly reduced in TRIM52-ΔLoop2 samples, BIRC6 association was not substantially changed (Fig. 5g, Supplementary Fig. 6i). These results are consistent with a model in which HUWE1 binds and ubiquitinates inside loop 2, whereas BIRC6 likely binds the TRIM52 RING outside of loop 2, yet ubiquitinates within it.

## BIRC6 and UBR4/KCMF1 ubiquitinate TRIM52 in vitro

To further investigate how the identified E3 ligases operate together to ubiquitinate TRIM52, we used in vitro ubiquitination assays. TwinStrep-EGFP-TRIM52 (TRIM52 hereafter) was expressed in insect cells and isolated by streptavidin affinity purification (Supplementary Fig. 7a). Recombinant TRIM52 protein auto-ubiquitinated and synthesized free poly-ubiquitin chains in in vitro reactions (Supplementary Fig. 7b), from which we concluded that TRIM52 itself has E3 ligase activity, despite its extended RING loop 2 region. Since auto-ubiquitination is dispensable for TRIM52 turn-over in cells[9], and the goal was to investigate TRIM52 as a substrate of the identified E3 ligases, we aimed to exclude ubiquitination by TRIM52 itself. We therefore expressed and purified a catalytically inactive TRIM52 allosteric linchpin[51] mutant (TRIM52-R187A), which no longer synthesized poly-ubiquitin chains (Supplementary Fig. 7c).

To test which E3 ligases could ubiquitinate TRIM52, and how many individual residues are modified within the substrate, we incubated TRIM52-R187A with the identified E3 ligases, ATP, and their cognate E1 and E2 enzymes. To assess the number of modified residues, we included lysine-less Dylight800-labeled ubiquitin (K0 ubiquitin), which prevents background signal from poly-ubiquitin chain formation. BIRC6 ubiquitinated TRIM52 on three residues in a time-dependent manner (Fig. 6a, and Supplementary Fig. 7d-e, Supplementary data 6), whereas UBR4/KCMF1 and HUWE1 did not efficiently ubiquitinate TRIM52-R187A (Fig. 6a). Similar reactions with wild-type ubiquitin yielded similar results (Fig. 6b), indicating that BIRC6 multi-mono-ubiquitinates full-length TRIM52. There was no evidence for efficient in vitro ubiquitination by UBR4/KCMF1 or HUWE1.

Cell-based experiments had indicated that the TRIM52 RING domain is sufficient as a substrate, and the loop 2 region is required for efficient ubiquitination (Fig. 5a-c). To test whether direct

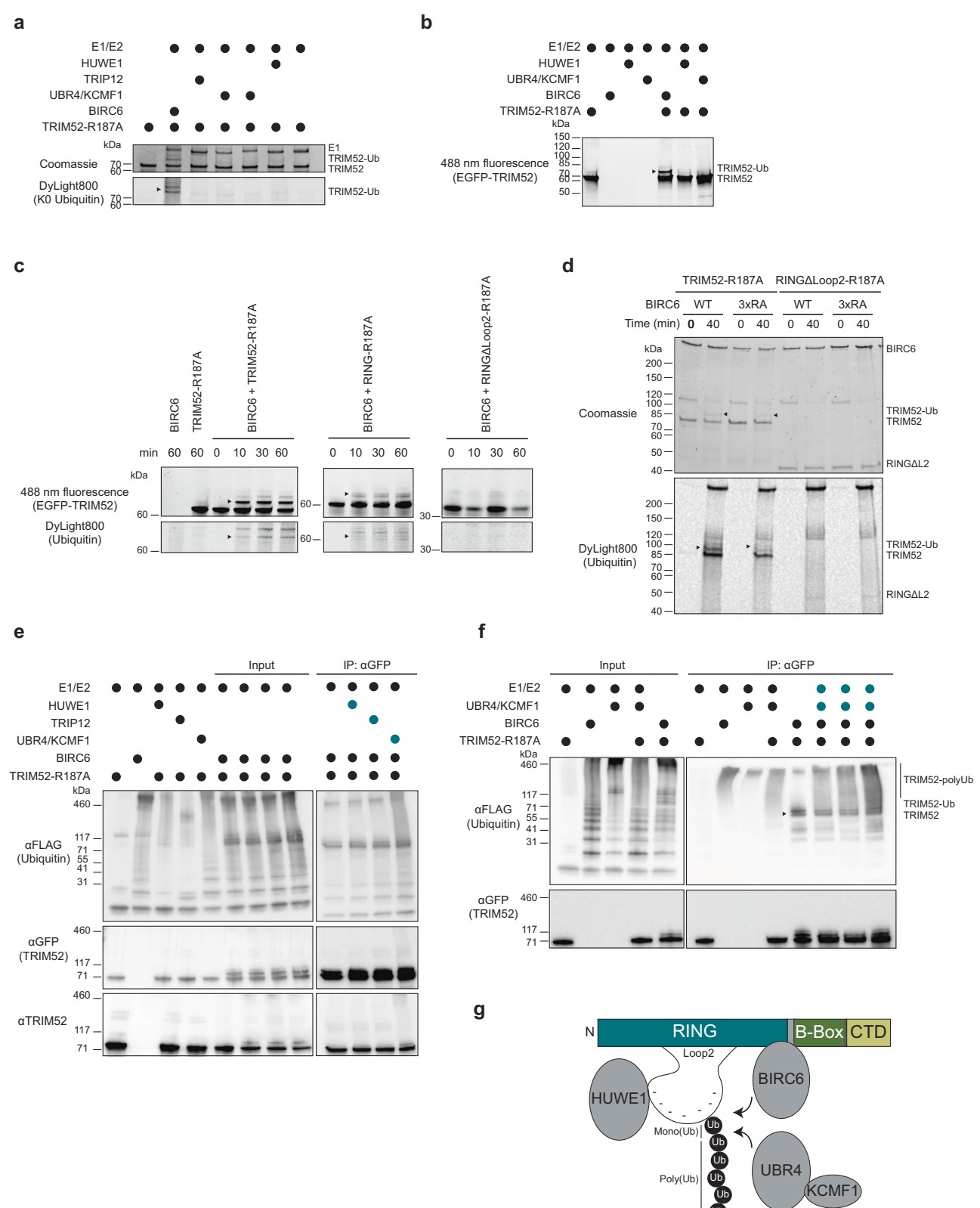

mono-ubiquitination by BIRC6 followed the same requirements, we used the recombinant RING domain (RING-R187A) and the RING domain lacking loop 2 (RING-ΔLoop2) as substrates in BIRC6-dependent ubiquitination assays. Consistent with cell-based results, the RING domain by itself was mono-ubiquitinated by BIRC6, while the RING mutant lacking the loop 2 region was not efficiently ubiquitinated

(Fig. 6c). While full-length TRIM52 was also ubiquitinated on two additional minor sites, the RING domain in isolation was not (Fig. 6c), suggesting that indeed the major ubiquitination site for BIRC6 is inside the TRIM52 RING domain. Although our co-IP data indicated that BIRC6 binds outside of loop 2, we hypothesized that its strong negative charge may still contribute to its recognition. A positively charged

**Fig. 6 | BIRC6 and UBR4/KCMF1 ubiquitinate TRIM52 in vitro. a** In vitro ubiquitination assay of purified EGFP-TRIM52-R187A with recombinant E3 ligases, their cognate E1 and E2 enzymes and DyLight800-labeled K0 ubiquitin. The reactions were incubated for 1 h. at 37 °C in the presence of ATP. **b** In vitro ubiquitination assay of EGFP-TRIM52-R187A with BIRC6 and in combination with HUWE1 or UBR4/KCMF1, their cognate E1 and E2s and WT ubiquitin. The reactions were incubated for 1 h. at 37 °C in the presence of ATP. **c** Time-course ubiquitination assay of EGFP-TRIM52-R187A, EGFP-RING-R187A, and EGFP-RING (ΔLoop2) with recombinant BIRC6, UBA6, and DyLight800-labeled ubiquitin. The reactions were incubated for indicated times at 37 °C in the presence of ATP. **d** Ubiquitination assay of EGFP-TRIM52-R187A, EGFP-RING (ΔLoop2) with BIRC6-WT or BIRC6-3xRA, UBA6 and DyLight800-labeled ubiquitin. The reactions were incubated for indicated times at 37 °C in the presence of ATP. **e** Ubiquitination assay coupled to

immunoprecipitation. Left: ubiquitination assays of EGFP-TRIM52-R187A with BIRC6, HUWE1, TRIP12, and UBR4/KCMF1, their cognate E1 and E2s as well as FLAG-labelled ubiquitin; right: αGFP IP of the indicated samples: the immunoprecipitated EGFP-TRIM52-R187A was further incubated with the indicated E3 ligases (labelled in teal), their cognate E1 and E2s as well as FLAG-ubiquitin. The reactions were incubated for 1 h. at 37 °C in the presence of ATP. **f** Ubiquitination assay coupled to immunoprecipitation. Input: ubiquitination assays of EGFP-TRIM52-R187A with BIRC6 or UBR4/KCMF1, their cognate E1 and E2s and FLAG-labelled ubiquitin. IP: αGFP IP of the indicated samples: the immunoprecipitated EGFP-TRIM52-R187A was further incubated in ubiquitination reactions with UBR4/KCMF1, their cognate E1 and E2s (labelled in teal) as well as FLAG-ubiquitin. The reactions were incubated for 1 h. at 37 °C in the presence of ATP. **g** Schematic model of TRIM52 degradation. Source data are provided as a Source Data file.

CBD-3 domain arginine loop at the centre of BIRC6 has been shown to mediate electrostatic interactions with a negatively charged region in two of its known substrates: HTRA2 and SMAC/DIABLO[52–54]. To test whether the same BIRC6 domain recognizes TRIM52, we used recombinant BIRC6 with a mutated CBD-3 arginine loop to negate its positive charge (3xRA)[52]. BIRC6-3xRA mono-ubiquitinated TRIM52 with a lower efficiency (Fig. 6d), indicating that the positive charge of the CBD-3 arginine loop region is partially required for the ubiquitination of TRIM52 by BIRC6.

BIRC6 mono-ubiquitinated TRIM52 in vitro, yet we identified abundant K48 poly-ubiquitin chains on TRIM52 isolated from cells (Fig. 3e-h, and Supplementary data 4), suggesting that these are the major degradation signals on TRIM52. Therefore, we asked whether UBR4/KCMF1 and HUWE1 can extend the BIRC6-seeded mono-ubiquitination into poly-ubiquitin chains. To this end, we mono-ubiquitinated TRIM52 in the presence of BIRC6, immunoprecipitated TRIM52, and then used it as an input substrate in ubiquitination reactions with the indicated E3 ligases (Fig. 6e). In the presence of UBR4/KCMF1, high molecular weight poly-ubiquitin species appeared, consistent with transition into poly-ubiquitin chains (Fig. 6e, f; three right-most samples represent replicates with different buffer conditions). A limited amount of ubiquitin signal below the MW of TRIM52 suggests that the samples contained a small amount of non-covalently bound free ubiquitin chains (Fig. 6e, f). However, the stronger ubiquitin signal above the TRIM52 MW suggested that a substantial fraction of the poly-ubiquitin chains was covalently attached to TRIM52. It should be noted that the limited amount of high molecular weight species in the GFP/TRIM52 blots indicates that under these reaction conditions, UBR4-dependent poly-ubiquitination of mono-ubiquitinated TRIM52 has limited efficiency in vitro. Together, these data suggest that TRIM52 is mono-ubiquitinated by BIRC6 on its RING domain, which can be extended by co-operation with UBR4/KCMF1 into poly-ubiquitin chains (Fig. 6g).

Previous reports indicate that BIRC6 localises in the cytoplasm and trans-Golgi network[55]. In contrast, our data indicate that TRIM52 is degraded in the nucleus, thus presenting the question how BIRC6-dependent ubiquitination of TRIM52 could contribute to its turn-over. To address this, RKO cells were fractionated, and BIRC6 analysed in the cytosolic and nuclear fractions. While we found BIRC6 to be predominantly present in the cytoplasm, we also detected a small fraction in the nucleus (Supplementary Fig. 7f). This suggests that there may be some nuclear BIRC6 that could directly mono-ubiquitinate TRIM52 in the nucleus, which subsequently facilitates its poly-ubiquitination and degradation. However, we cannot exclude the alternative option that TRIM52 is mono-ubiquitinated by BIRC6 in the cytoplasm, yet poly-ubiquitinated and degraded after translocation to the nucleus.

## Discussion

TRIM52 has been under positive selection pressure in humans and old-world primates. In contrast to other TRIM proteins with similar

evolutionary patterns -like TRIM5α- no evidence has been found for direct anti-retroviral activity of TRIM52[6], although some studies have reported a role in defences against other viruses and inflammatory signaling[10,56–58]. In agreement with cell-intrinsic selection pressure, rather than pressure exclusively from exogenous pathogens, we found that *TRIM52* ablation decreased cellular fitness in a p53-dependent manner, in the absence of external stimuli[8,9]. This suggested a role of TRIM52 in maintaining primate genome integrity. Like *TRIM52*, many other regulators of the DNA damage response have been under positive selection pressure in humans and other primates[1,2], although the selective advantages of these adaptations have remained poorly defined.

Here, we found that TRIM52 closely associates with TDP2 and ZNF451, known regulators of topoisomerase-dependent DNA lesions. Such lesions would be predominantly resolved through HDR and error-free NHEJ, yet can in part be compensated for by error-prone NHEJ[36,59,60]. Our genetic modifier screen showed that loss of key components of NHEJ repair are synthetically lethal with loss of *TRIM52*, pointing to a physiologic role of TRIM52 in limiting the accumulation of topoisomerase lesions, and ultimate activation of DSB repair pathways.

*TRIM52* ablation increased the concentrations of covalently-associated DNA-TOP2 complexes, indicating that TRIM52 is required for limiting the accumulation of irreversible TOP2-DNA tyrosyl cleavage complexes. Removal of the TOP2ccs from the DNA can be mediated through proteasomal degradation, or in a proteasome-independent manner[28–30,32]. It should be noted that while loss of *TRIM52* in the absence of etoposide showed a consistent and reproducible trend towards increased DNA damage markers (Fig. 2d-h), these changes were non-significant. We speculate that a combination of the effect size, experimental variance, and/or number of replicates may underlie the limited statistical power in these samples. Nevertheless, low increases in DNA damage from endogenous sources such as genome replication or transcription are consistent with relatively slow p53 activation and cell cycle arrest upon TRIM52 loss[8]. In the *TRIM52* double-knock-out experiments with key components of the NHEJ and HDR branches (Supplementary Fig. 1a-b) the effect on cell fitness was measured. From these experiments, we concluded that both NHEJ and HDR are functionally redundant for TRIM52-dependent cellular output. These results are consistent with cell-fitness loss stemming from DNA damage in the *TRIM52* knock-out cells. However, further future examination of DNA damage markers in non-stimulated *TRIM52* knock-out cells will be important to further substantiate this conclusion.

*TRIM52* arose by partial gene duplication of the evolutionary conserved *TRIM41* gene, which has been implicated in the ubiquitination of TOP3B cleavage complexes and their subsequent proteasomal degradation[61], suggesting that *TRIM52* divergent evolution may have resulted in an activity for TOP2, although there is currently no experimental evidence for this. The precise stage and mechanism of

limiting the accumulation of TOP2cc complexes by TRIM52, its crosstalk with TRIM41, and its effects on other topoisomerases will require additional future study.

RING E3 ligases interact with UBC folds in E2 conjugase enzymes, in part through interactions involving the two protruding RING loops[62]. Despite its disproportionately large and charge-biased loop 2, we unexpectedly found that TRIM52 had E3 ligase activity in vitro in combination with the promiscuous UBCH5B/C E2 conjugases. Especially since catalytic activity is not required for its own degradation[9], it is thus tempting to speculate that TRIM52's E3 activity may be required for its cellular function in limiting topoisomerase-mediated lesions.

Although we identified TRIM52 to be part of TDP2-containing complexes in cells, their individual ablation had different effects on TOP2cc levels, suggesting that these factors may function in parallel to each other. Consistent with previous findings in other cell models, *TDP2* knock-out alone had no measurable effect on TOP2cc levels on the DNA[25], whereas these were increased in cells lacking *TRIM52*. In combination with TRIM52's rapid proteasomal degradation, these findings suggest that TRIM52 may regulate the resolution of TOP2 lesions. Although additional studies will be required to determine the relationship between TRIM52 degradation and its cellular function, we speculate that TRIM52 is either constantly degraded, yet stabilized when engaged in a DNA repair complex, or it is actively degraded when engaged in such complexes.

UBR4 operates with the UBE2A (RAD6) E2 conjugase enzyme[63], which was identified alongside UBR4 and KCMF1 in our genetic screen. KCMF1 has been previously described as an interactor and co-factor of UBR4, although its function has remained enigmatic[46,64–67]. UBR4 was active in vitro in the absence of KCMF1, as it produced free ubiquitin chains, whereas KCMF1 by itself showed no E3 ligase activity. This suggests that KCMF1 could be a modular co-factor of UBR4, conferring recognition of mono-ubiquitinated substrates and facilitating E4 ubiquitin chain-extending activity to UBR4, or stabilizing the UBR4 protein itself. UBR4 and KCMF1 have been implemented in vesicular trafficking and autophagosomal/lysosomal protein degradation[46,64,68], which our inhibitor studies indicate play no role in TRIM52 degradation. This raises the question how cells distinguish the ultimate destination of degradation for UBR4/KCMF1-dependent substrates.

The three giant E3 ligase components -HUWE1[48,69–75], BIRC6[53,55,65,76,77], and UBR4/KCMF1[46,68,78–85]- we identified to degrade TRIM52 have been previously implicated in the degradation of a wide range of substrates, thereby regulating various cellular processes. This has suggested that these E3 ligases can recognize multiple substrate classes based on wider biochemical or biophysical substrate features. Consistent with this notion, recent elucidation of the structure of HUWE1[49,50] showed that this E3 ligase has three substrate binding modules allowing the recognition and ubiquitination of non-engaged nucleic acid-binding proteins and ubiquitinated/PARylated substrates. Likewise, BIRC6 binds and ubiquitinates various different apoptosis- and autophagy-related proteins[52,53,86]. Although such information is missing for UBR4 as its full-length structure has not been published, the fact that UBR4 has been found in independent screens to ubiquitinate aggregation-prone nascent polypeptides during proteotoxic stress[84], various mitochondrial proteins[80,83], and ER-associated degradation substrates[85], indicates that also this giant E3 ligase may recognize various classes of substrates, potentially dependent on its association with different partners, such as KCMF1 and Calmodulin[46,87].

Despite the fundamental insight that these E3s can recognize multiple substrate classes, bona fide cellular substrates and how they are recognized have remained sparse. This study has identified TRIM52 as a BIRC6 substrate for mono-ubiquitination. Our cellular mutagenesis data indicate that BIRC6 likely recognizes and binds the RING domain. This raises the question whether any of the other -600 RING domain proteins are likewise BIRC6 substrates. Moreover, BIRC6 had so far been shown to poly-ubiquitinate its substrates[52,86], meaning it

can both recognize substrates, modify them with the first ubiquitin, and extend this PTM into a poly-ubiquitin chain. While we cannot rule out that BIRC6 performs all of these actions on TRIM52 in cells in the presence of co-factors or PTMs, in vitro BIRC6 performed exclusively TRIM52 multi-mono-ubiquitination. This positions TRIM52 as an excellent substrate for future endeavours to elucidate how BIRC6 mono- and poly-ubiquitination activities are mechanistically controlled.

The fact that UBR4/KCMF1 and HUWE1 have been repeatedly identified to regulate partially overlapping substrates in cells could indicate the existence of a large protein complex consisting of some or all of the identified E3 ligases. In this context, cooperative action of BIRC6 and UBR4/KCMF1 controls activation of the Integrated Stress Response by ubiquitination of heme-regulated inhibitor (HRI)[65]. Therefore, like TRIM52, other cellular substrates may rely on sequential mono-ubiquitination by BIRC6, followed by UBR4/KCMF1 chain extension.

In summary, we identified a functional role of TRIM52 in maintaining genome integrity despite its non-conserved nature across mammals. These findings will enable future studies to identify what evolutionary benefit TRIM52 provides for humans, and perhaps brain complexity. In addition, we unravelled how one of the most unstable human proteins is degraded. We identified TRIM52 as a bona fide substrate for BIRC6 and UBR4/KCMF1, requiring its acidic loop 2 for recognition and degradation. While HUWE1 is functionally important for TRIM52 turn-over, its mechanistic targeting of TRIM52 will need further clarification in future studies. Together, these findings form the basis for follow-up studies addressing how TRIM52 turn-over is related to its cellular function.

## Methods
### Cell culture
HEK-293T cells were cultured in high glucose Dulbecco's Modified Eagle's Medium (DMEM; Sigma-Aldrich, D6429) supplemented with 10% Fetal Bovine Serum (FCS; Sigma-Aldrich, F7524) and 1% Penicillin-Streptomycin (Sigma-Aldrich, P4333). RKO cells were cultured in Roswell Park Memorial Institute 1640 Medium (RPMI; Thermo Fisher Scientific, 21875) supplemented with 10% FCS (Sigma-Aldrich, F7524), 2% sodium pyruvate (Sigma-Aldrich, S8636), 1% Minimum Essential Medium (MEM) Non-Essential Amino Acids (Thermo Fisher Scientific, 11140050), and 1% Penicillin-Streptomycin (Sigma-Aldrich, P4333). All cells were cultured at 37 °C and 5% $CO_2$ in a humidified incubator. Cells were treated with the following reagents for the indicated times: 200 µg/ml cycloheximide (CHX; Sigma-Aldrich, C1988); 10 µM MG132 (Sigma-Aldrich, M7449); 10 µM epoxomicin (Gentaur Molecular Products, 607-A2606); 200 ng/ml doxycycline hyclate (Dox; Sigma-Aldrich, D9891); 0.5-1 mg/ml G418 (Sigma-Aldrich, A1720); 400 nM bafilomycin A1 (Santa Cruz Biotechnology, sc-201550); 20 µM MLN-4924 (Abcam, ab216470), 5 µM etoposide (ETO, Sigma-Aldrich, E1383), 5 µM capzimin[40]. For gene targeting, HEK-293T cells were transduced with LentiCRISPR_V2 vectors (Addgene plasmid 52961; http://n2t.net/addgene:52961; RRID: Addgene_52961) encoding the indicated sgRNA sequences. Transduced cells were selected by supplementing DMEM culture media with 4 µg/ml puromycin (Invivogen, ant-pr-1). In RKO-Cas9 cells genome editing was induced using 200 ng/ml or 350 ng/ml final concentration of doxycycline hyclate (Dox, Sigma-Aldrich, D9891). Cas9 genome editing and expression in the absence of Dox from the TRE3G promoter was tested with competitive proliferation assays[8]. The cell lines, culture conditions and reagents used in this study are listed in the supplementary information. Cell lines used in this study were authenticated by STR analysis, and tested for mycoplasma contamination.

### Vectors
The lentiviral human sgRNA library was designed to targeted ubiquitin-proteasomal system-related genes, and is made up of 6 sgRNAs

targeting each of the selected genes[19]. Lentiviral vectors expressing sgRNAs under a U6 promoter as well as selection colors EBFP or iRFP from a PGK promoter have been previously described[20]. sgRNA CDSs were cloned into pLentiv2-U6-PGK-iRFP670-P2A-Neo[20] and used for gene targeting in RKO cell lines. The TRIM52 stability reporter (pLX-SFFV-MYC-mCherry-P2A-OLLAS-EGFP-TRIM52) was designed by cloning the open reading frame of human *TRIM52* into a modified pLX303 vector[47,88]. Single sgRNA CDSs were cloned in pLentiCRISPRv2 (Addgene plasmid 52961) to perform stable knock-outs in HEK-293T cells. cDNAs encoding TRIM52 FL, TRIM52 (KtoR), TRIM52 (2KtoR), TRIM52 (5KtoR), TRIM52 (ΔLoop2), RING, RING (ΔLoop2), RING (Loop2-neut), RING (Loop2-basic), DAXX (411-559), CY2B (1-103) were purchased from Twist Bioscience or generated by strand overlap PCR, and cloned into a modified pLX303 vector[88]. The plasmids and sgRNAs used in this study are listed in the supplementary information.

## Transfection of HEK-293T cells and production of virus-like particles

Transfection mixes were made containing DNA and polyethylenimine (PEI; Polysciences, 23966) in a ratio of 1:3 (μg DNA/μg PEI) in DMEM (Sigma-Aldrich, D6429) without supplements. The day prior to transfection, HEK-293T cells were seeded in 6-well clusters in supplemented DMEM media. For virus production, transfections were performed using 500 ng psPAX2 (Gag-Pol) plasmid (Addgene plasmid 12260; http://n2t.net/addgene:12260; RRID:Addgene_12260), 500 ng mini-genome plasmid and 100 ng pCMV2-VSVG plasmid[89] in 6-well clusters. Transfected cells were incubated for 72 h. at 37 °C, after which virus-like particles were harvested by filtering the supernatant through a 0.45 μm filter. Virus-like particles were directly used after harvesting, or kept at 4 °C for short-term storage and at -80 °C for long-term storage.

## Cell competition assays

Competitive cell fitness assays were performed as described previously[90]. In brief, RKO cells harbouring Dox-inducible Cas9 were transduced with iRFP or EBFP lentiviral sgRNA plasmids targeting the indicated genes. The multiplicity of transduction was such that 30-60% of cells were iRFP/EBFP-positive before the start of fitness measurements. Gene editing was induced with 200 ng/ml final concentration of Dox (Sigma-Aldrich, D9891) and the percentage of iRFP/EBFP-positive cells monitored for twenty days by flow cytometry at the indicated days. The relative fraction of sgRNA-positive cells was normalized to sg*hROSA* or sg*AAVS1* of the same cell line on day 0.

## CRISPR-iCas9-based modifier screens

RKO-iCas9-P2A-BFP cells carrying a genome-wide sgRNA library[19] were grown in G418- (0.5 mg/ml, Sigma-Aldrich, A1720) and Dox-containing (200 ng/ml, Sigma-Aldrich, D9891) RPMI medium for iCas9 mediated base drop out of essential genes (T0). On day 5, cells were further transduced with VLPs for expression of a pair of sgRNAs targeting either *TRIM52*, *DGCR8* or the control locus *AAVS1* and selected in puromycin- (1 μg/ml) and Dox-supplemented RPMI. Cells were passaged for 12 doublings of control sgRNA (AAVS1) expressing cells (T3). All cells were grown at 500-fold library representation. The fraction of library-positive cells was monitored regularly by Thy1.1 surface marker staining and flow cytometry. Samples for NGS sequencing were collected at timepoints T0 and T3. For harvesting, $1.5 \times 10^8$ cells were pelleted, washed with PBS, and stored at -80 °C until further processing.

## FACS-based CRISPR–iCas9 genetic screens

Lentivirus-like particles were used to transduce RKO-MYC-mCherry-P2A-OLLAS-EGFP-TRIM52 cells at a multiplicity of infection (MOI) of less than 0.2 TU/cell, and 1000-fold library representation. The percentage of library-positive cells was determined

after 4 days of transduction by immunostaining of the Thy1.1 surface marker and subsequent flow cytometric analysis. RKO cells with integrated lentiviral vectors were selected with G418 (1 mg/ml, Sigma-Aldrich, A1720) for 5 days, after which they were maintained in 0.5 mg/ml G418. After G418 selection, 20 million cells of unsorted reference samples were collected and stored at −80 °C until further processing. Cas9 genome editing was induced with Dox (200 ng/ml, Sigma-Aldrich, D9891) and after 3 and 6 days, cells were sorted by FACS. Cells were harvested, washed with PBS and sorted in fully supplemented RPMI-1640 using the FACS Aria III cell sorter operated by BD FACSDiva software (v8.0). RKO cells were gated for live, single, EBFP-positive (Cas9 expression), EGFP-positive, mCherry-positive, and. 2-3% of cells with the lowest and 2-4% of cells with the highest EGFP or mCherry signals were sorted into PBS. At least $1 \times 10^6$ (EGFP$^{low}$ and mCherry$^{low}$) and $1 \times 10^6$ (EGFP$^{high}$ and mCherry$^{high}$) cells were collected for each time point. Sorted samples were re-analysed for purity, pelleted and stored at −80 °C until further processing. The gating strategy for flow cytometric cell sorting is shown in the Supplementary Information.

## Next-generation sequencing library preparation and genetic screen analysis

Next-generation sequencing (NGS) libraries of sorted and unsorted control samples were processed as previously described[20]. In brief, isolated genomic DNA was amplified with two-step PCR. The first PCR amplified the integrated sgRNA cassettes, and the second PCR introduced the Illumina adapters. Purified PCR products' size distribution and concentrations were measured using a fragment analyser (Advanced Analytical Technologies). Equimolar ratios of the obtained libraries were pooled and sequenced on a HiSeq 2500 platform (Illumina). Primers used for library amplification are listed in the supplementary information. Analysis of the CRISPR–Cas9 screen was performed as previously described[20]. In brief, sgRNAs enriched in day 3 and day 6 post-Cas9 induction-sorted samples were compared against the matching unsorted control populations harvested on the same days using MAGeCK[91].

## RADAR assay

RADAR assays were performed as previously described[39]. In brief, $1 \times 10^6$ RKO Cas9 cells were treated with etoposide (10 μg/ml) for 2 h., washed with PBS and lysed by adding 1 mL DNAzol (Thermo Fisher Scientific, 11558626). Nucleic acids were precipitated by the addition of 0.5 mL 100% ethanol, incubation at −20 °C for 5 min., and centrifuged at 12,000 × *g* for 10 min. Precipitated nucleic acids were washed twice by addition of 75% ethanol and vortexing, resuspended in 200 μl TE buffer and heated at 65 °C for 15 min. DNA was sheared by sonication (40% power for 15 s. pulse and 30 s. rest 5 times). Samples were centrifuged at 20,000 × *g* for 5 min. and supernatant-containing nucleic acids with covalently bound protein collected. Double-stranded DNA content of the sample was quantified by PicoGreen assay kit (Invitrogen, 11558626) and slot-blotted. TOP2cc were detected with anti-TOP2α antibody.

## Protein half-life determination

To estimate TRIM52 protein half-lives, RKO cell lines stably expressing MYC-mCherry-P2A-3xHA-TRIM52 or MYC-mCherry-P2A-3xHA-TRIM52-KtoR were treated with 200 μg/ml of cycloheximide (CHX, Sigma-Aldrich, C1988). At indicated time points, total protein extracts were generated using 1x disruption buffer (1.05 M Urea, 0.334 M β-Mercaptoethanol and 0.7% SDS) analysed by WB, quantified, and normalized to stable internal control MYC-mCherry levels, and to time point 0 as indicated. Single exponential decay curves were plotted using GraphPad Prism (v9), from which protein half-lives were calculated.

## Co-immunoprecipitation assays

HEK-293T cells from one confluent 35-mm dish were lysed in 100 µl of Frackelton lysis buffer (10 mM Tris (pH 7.4), 50 mM NaCl, 30 mM Na$_4$P$_2$O$_7$, 50 mM NaF, 2 mM EDTA, 1% Triton X-100, 1 mM DTT, 1 mM PMSF, and 1X protease inhibitor cocktail (cOmplete™ Protease Inhibitor Cocktail, 11697498001)). Cells were incubated on a rotating wheel at 4 °C for 30 min. then centrifuged at 20,000 × *g* at 4 °C for 30 min. Supernatants were transferred to new tubes and 10 µl (10% of the lysate) of each sample was collected as input fractions. Protein concentrations were determined by BCA Protein Assay Kit (Thermo Fisher Scientific, 23225) and 500 µg of lysates were incubated overnight at 4 °C on a rotating wheel with anti-HA antibody (Cell Signaling Technology, 1:100). The next day, magnetic beads (Protein A/G Magnetic Beads, Thermo Fisher Scientific, 88803), were blocked by rotation in 3% BSA in Frackelton Buffer for 1 h. at 4 °C. 25 µl of beads were added to 500 µg of lysates and rotated for 2 h. at 4 °C. Then, beads were washed five times with 1 ml of Frackelton buffer, and proteins eluted by boiling in 2X disruption buffer (2.1 M urea, 667 mM β-mercaptoethanol and 1.4% SDS) for 10 min. at 95 °C.

## Immunoprecipitations for ubiquitination

RKO cells stably expressing MYC-mCherry, MYC-mCherry-TRIM52 or MYC-mCherry-P2A-3xHA-TRIM52 were treated with 10 µM epoxomicin for 5 h. Cells from one confluent 35-mm dish were lysed in 100 µl of RIPA buffer with 1% SDS (50 mM Tris-HCl (pH 7.4), 150 mM NaCl, 1% SDS, 0.5% sodium deoxycholate, 1% Triton X-100), supplemented with 40 mM N-ethylmaleimide, 40 mM iodoacetamide, 25 U/ml benzonase, 1 mM PMSF, and 1X protease inhibitor cocktail (cOmplete™ Protease Inhibitor Cocktail, 11697498001). Cells were incubated on a rotating wheel at 4 °C for 30 min., and centrifuged at 20,000× *g* at 4 °C for 15 min. Supernatants were transferred to new tubes. Protein concentrations were determined by BCA Protein Assay Kit (Thermo Fisher Scientific, 23225), and 30 µg of the lysates were collected as input. 500 µg of lysates were diluted 1:10 in RIPA buffer without SDS (50 mM Tris-HCl (pH 7.4), 150 mM NaCl, 0.5% sodium deoxycholate, 1% Triton X-100) and incubated with anti-HA antibody (Cell Signaling Technology, 1:100) overnight, or 25 µl of magnetic beads (RFP-Trap Dynabeads, Chromotek, rtd-20) for 2 h. Prior to incubation, beads were blocked by rotation in 3% BSA in RIPA Buffer for 1 h. at 4 °C. For the HA-IP, the next day 25 µl of magnetic beads (Protein A/G Magnetic Beads, Thermo Fisher Scientific, 88803) were added to 500 µg of lysates and rotated for 2 h. at 4 °C. Subsequently, beads were washed five times with 1 ml of RIPA buffer, supplemented with 600 or 300 mM NaCl respectively. Proteins were eluted by boiling in 2X disruption buffer (2.1 M urea, 667 mM β-mercaptoethanol and 1.4% SDS) for 10 min. at 95 °C.

## Immunoprecipitations for ubiquitination site identification by nLC-MS/MS

HEK 293 T cells were transfected with MYC-mCherry-TRIM52 and 6xHis-Ubiquitin expression plasmids and after 48 h. treated with 10 µM epoxomicin (Gentaur Molecular Products, 607-A2606) for 5 h. Cells from 5 confluent 15-cm dishes were lysed in 1 ml of RIPA buffer with 1% SDS (50 mM Tris-HCl (pH 7.4), 150 mM NaCl, 1% SDS, 0.5% sodium deoxycholate, 1% Triton X-100), supplemented with 40 mM N-ethylmaleimide, 40 mM iodoacetamide, 25 U/ml benzonase, 1 mM PMSF, and 1X protease inhibitor cocktail (cOmplete™ Protease Inhibitor Cocktail, 11697498001). Lysates were incubated on a rotating wheel at 4 °C for 30 min., and centrifuged at 20,000 x g at 4 °C for 30 min. Supernatants were transferred to new tubes. Protein concentrations were determined by BCA Protein Assay Kit (Thermo Fisher Scientific, 23225), and 30 µg of lysate was collected as input. The rest of the lysates were diluted 1:10 in RIPA no-SDS (50 mM Tris-HCl (pH 7.4), 150 mM NaCl, 0.5% sodium deoxycholate, 1% Triton X-100) and incubated with 250 µl of magnetic beads (RFP-Trap Dynabeads, Chromotek, rtd-20) for 2 h. Subsequently, beads were washed five times with

1 ml of RIPA wash buffer (50 mM Tris-HCl (pH 7.4), 600 mM NaCl, 0.1% SDS, 0.5% sodium deoxycholate, 1% Triton X-100) supplemented with 600 mM NaCl. Protein was eluted from the beads in 1 ml elution buffer (50 mM Tris-HCl (pH 8), 150 mM NaCl, 0.1% SDS, 4 M urea, 5 mM imidazole), supplemented with 40 mM N-ethylmaleimide, 40 mM iodoacetamide. Samples were then diluted 1:10 in wash buffer (50 mM Tris-HCl (pH 8), 150 mM NaCl, 4 M urea, 5 mM imidazole) and incubated with 500 µl Ni-NTA beads (Thermo Fisher Scientific, 88831) at 4 °C for 2 h. Beads were washed 5 times in 1 ml of wash buffer on a rotating wheel at 4 °C for 5 min. After washes, beads were re-suspended in 100 µl 50 mM Tris-HCl (pH 8) and analysed by nLC-MS/MS.

## TurboID proximity labelling

The TurboID sequence from plasmid V5-TurboID-NES_pCDNA3[24] (Addgene plasmid: 107169) was PCR amplified and integrated into a modified version of pCW57.1[88]. TRIM52 and EGFP were cloned into this plasmid for their expression with an N-terminal TurboID tag. Subsequently, RKO cell lines stably expressing various Dox-inducible TurboID constructs were created. For the TurboID experiments cell lines expressing Dox-inducible TurboID-TRIM52 and TurboID-EGFP were stimulated with 55 or 250 ng/ml doxycycline hyclate (Dox; Sigma-Aldrich, D9891) for 48 h. to induce the expression of the TurboID fusion constructs. Subsequently, cells were stimulated for 4 h. with 10 µM epoxomicin (Gentaur Molecular Products, 607-A2606) to inhibit proteasomal degradation where indicated or left untreated, and finally, biotinylation was induced for 15 min. by addition of 500 µM biotin (Sigma, B4501) to the cell culture medium. Biotinylated proteins were enriched with streptavidin-coated beads. In brief, cells were washed four times with ice-cold PBS, prior to lysis in RIPA lysis buffer (50 mM Tris HCl (pH 7.4), 150 mM NaCl, 1% NP-40, 0.5% Sodium Deoxycholate, 1 mM EDTA, 0.1% SDS, 1 mM PMSF and 1X protease inhibitor (cOmplete™ Protease Inhibitor Cocktail, 11697498001)). Lysates were rotated for 15 min. at 4 °C, centrifuged at 18,500x *g* for 10 min. at 4 °C, and protein concentrations were determined by BCA assay. 1200 µg of protein were incubated overnight rotating at 4 °C with 200 µl of streptavidin beads (Thermo Scientific, 88816), which were acetylated with Sulfo-NHS-Acetate beforehand, as described[92]. Beads were washed twice with 1 ml of RIPA buffer, once with 1 ml of 2 M Urea in 10 mM Tris (pH 8), twice with 1 ml RIPA buffer and five times with 50 mM HEPES (pH 7.8). Three technical replicates were subjected to nLC-MS/MS analysis.

For nLC-MS/MS sample preparation, the beads were resuspended in 40 µl 1 M urea and 50 mM ammonium bicarbonate. Disulfide bonds were reduced with 1.6 µl of 250 mM dithiothreitol (DTT) for 30 min. at RT before adding 1.6 µl of 500 mM iodoacetamide and incubating for 30 min. at RT in the dark. The remaining iodoacetamide was quenched with 0.8 µl of 250 mM DTT for 10 min. Proteins were digested with 150 ng LysC (mass spectrometry grade, FUJIFILM Wako chemicals) at 25 °C overnight. The supernatant was transferred to a new tube and digested with 150 ng trypsin (Trypsin Gold, Promega) in 1.5 µl 50 mM ammonium bicarbonate at 37 °C for 5 h. The digest was stopped by the addition of trifluoroacetic acid (TFA) to a final concentration of 0.5 %, and the peptides were desalted using C18 Stagetips[93].

## Co-immunoprecipitation of purified proteins

1 µg of purified EGFP-TRIM52 (WT or domain mutants) and 1 µg of purified HUWE1 were incubated in ubiquitination buffer (25 mM HEPES (pH 7.5), 150 mM KCl, 4 mM MgCl$_2$, 0.5 mM TCEP) supplemented with 1x protease inhibitor cocktail (cOmplete™ Protease Inhibitor Cocktail, 11697498001) for 2 h. at 4 °C on a rotating wheel. In the meantime, GFP-Trap magnetic beads (Chromotek, gtd-20) were blocked by rotation in 3% BSA ubiquitination buffer for 1 h. at 4 °C. After the incubation, 5% of the reaction volume was aliquoted for the input fraction. The rest of the sample was incubated with 10 µl magnetic beads for 30 min. at 4 °C on a rotating wheel. Beads were washed

5 times with 500 µl ubiquitination reaction buffer. Proteins were eluted by boiling in 2X disruption buffer (2.1 M urea, 667 mM β-mercaptoethanol and 1.4% SDS) for 10 min. at 95 °C and analysed by SDS-PAGE and WB.

## In vitro ubiquitination assays

In vitro ubiquitination assays contained 20 µM ubiquitin (WT, K0 or FLAG-tagged), 1 µM DyLight800/FLAG-labelled ubiquitin for in-gel visualization, 0.2 µM E1 (UBA6, UBA1), 0.4 µM E2 (RAD6, UBCH5B) 0.4 µM E3 (BIRC6, UBR4, HUWE1, KCMF1) and 2 µM substrate protein (EGFP-TRIM52), unless indicated otherwise. The assays were performed in 25 mM HEPES pH 7.5, 150 mM KCl, 4 mM MgCl$_2$, 0.5 mM TCEP (assay buffer) at 37 °C for the indicated times in 10-20 µl volumes. Reactions were initiated by the addition of 5 mM ATP and stopped by adding 2x reducing SDS-PAGE loading buffer. SDS-PAGE was performed using 4–20% Mini-PROTEAN TGX Stain-Free (BioRad, 4568094) gels. A Bio-Rad ChemiDoc MP system was used for in gel fluorescence imaging.

## Immunofluorescence confocal microscopy

RKO cells stably expressing mCherry-TRIM52, EGFP-TRIM52, TurboID-TRIM52 or TurboID-EGFP were seeded onto coverslips. TurboID-TRIM52 or TurboID-EGFP expressing cells were treated with Dox (1 µg/ml) for 4 days to induce TurboID expression. After 48 h., cells were treated with 10 µM epoxomicin (Gentaur Molecular Products, 607-A2606) for 5 h. Cells were washed once with PBS, fixed on coverslips with 4% PFA for 15 min. at RT and washed twice with PBS for 5 min. Cells were permeabilized with PBS-0.25% Triton X-100 for 5 min. at RT and washed three times with PBS before blocking for 30 min. in blocking solution (PBS, 1% BSA). TurboID-TRIM52 or TurboID-EGFP cells were stained with anti-MYC antibody (4A6, 1:500 in 1% BSA) for 1 h. at 37 °C, in a moisture chamber, washed three times with PBS and subsequently stained with AF647-labeled anti-mouse secondary antibody (1:200, Abcam, ab169348 in 1% BSA) for 1 h at RT, protected from light. Cells were washed three times with PBS and incubated with 0.4X Hoechst (Thermo Fisher Scientific) in PBS. The coverslips were mounted using ProLong™ Gold Antifade Mountant (Invitrogen). Images were collected using a Zeiss LSM 980 confocal microscope at 40X magnification.

## γH2AX staining

RKO cells harbouring Dox-inducible Cas9 transduced lentiviral sgRNA plasmids targeting *TRIM52* or *AAVS1*. Gene editing was induced with 200 ng/ml final concentration of Dox (Sigma-Aldrich, D9891). 96 h. post-Cas9 induction cells were seeded onto coverslips. After 48 h., cells were treated with etoposide (10 µg/ml) or DMSO for 2 h., Cells were washed once with PBS, fixed on coverslips with 4% PFA for 10 min. at RT and washed twice with PBS twice. Cells were permeabilized with PBS-0.3% Triton X -100 (pH = 7.4) and washed twice with PBS before blocking for 1 h. in blocking solution (PBA, 3% BSA, 10% FBS (pH = 7.4)) at RT. Cells were stained with anti-γH2AX antibody (A300-083, Bethyl, 1:2000 in blocking solution) overnight at 4 °C in a moisture chamber, washed twice with PBS for and subsequently stained with AF594 anti-rabbit secondary antibody (Invitrogen, A11012, 1:1000 in blocking solution) for 1 h at RT, protected from light. Cells were washed twice with PBS, once in sterilized ddH2O and mounted using ProLong™ Diamond Antifade Mountant with DAPI (Invitrogen).

## Subcellular fractionation

Subcellular fractionation was performed as previously described[94]. Two million cells were washed in 1 ml PBS and lysed in 500 µl ice-cold REAP buffer (0.1% NP-40 in 1x PBS) supplemented with 1 mM PMSF (Sigma-Aldrich), 1X cOmplete Protease Inhibitor Cocktail (Roche) and Benzonase. 240 µl of the lysates were collected as

whole cell fractions, the remaining lysate was centrifuged at 3000x *g* for 60 s. at 4 °C. 240 µl of supernatants were collected as cytosolic fractions, after which pellets were washed with 500 µl of REAP buffer, collected by centrifugation at 3000x *g* for 60 s. at 4 °C, and then resuspended in 240 µl of REAP buffer (nuclear fraction). 6x Laemmli sample buffer was added to all fractions (62.5 mM Tris-HCl (pH 6.8), 5.8% Glycerol, 2% SDS and 1.7% β- Mercaptoethanol), and boiled at 95 °C for 10 min. Equal volumes of fractions were loaded on a 10 % SDS polyacrylamide gel.

## Statistics and Reproducibility

The experiment represented in Fig. 2g, h was performed once, in technical triplicates. Identification of ubiquitination sites by nLC-MS/MS represented in Fig. 3g, h, h was performed once on a single sample. The experiments in Figs. 3a, e, f, 3i, 5b, f-h, 6a-f were repeated twice with similar results. The experiments in Supplementary Figs. 6d, g, 7e were performed once. The experiments in Supplementary Figs. 2b-c, 4d, 5c, 6c, f, i, 7a-d were repeated twice with similar results. The experiments in Supplementary Figs. 2g, 4b, 7f were repeated three times with similar results.

## Reporting summary

Further information on research design is available in the Nature Portfolio Reporting Summary linked to this article.

## Data availability

The mass-spectrometry data generated in this study have been deposited at the ProteomeXchange Consortium via the PRIDE partner repository with the data set identifier PXD051295 and PXD051272. All remaining data generated or analysed during this study are included in the manuscript and its supporting files. The genetic screen data generated in this study are provided in the Source Data file. Source data are provided with this paper.

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

## Acknowledgements

Next Generation Sequencing analysis Vienna Biocenter Core Facilities (VBCF) was performed by the Vienna Biocenter Core Facilities using the VBCF instrument pool. Proteomics analyses were performed by the Mass-spectrometry Facility at the Max Perutz Labs using the Max Perutz Labs and VBCF instrument pool; in particular, we thank Markus Hartl and WeiQiang Chen for their expert support. Flow cytometry analyses were performed at the BioOptics FACS Facility at Max Perutz Labs

using the Max Perutz Labs instrument pool; in particular, we acknowledge Kitti Csalyi, Thomas Sauer and Johanna Stranner for expert support. Microscopy was performed at the BioOptics Light Microscopy Facility at the Max Perutz Labs; we thank Thomas Peterbauer and Irmgard Fischer for their expert support and training. We are grateful to Robert Kurzbauer and Maria Heinke for the purification of recombinant proteins. We thank Life Science Editors for editing services. This research was funded in whole, or in part, by the Austrian Science Fund (FWF) (grants 10.55776/P36572, 10.55776/P30415, 10.55776/P30231, 10.55776/P36945, 10.55776/F79, and 10.55776/W1261 to G.A.V.). For the purpose of open access, the author has applied a CC-BY public copyright licence to any Author Accepted Manuscript version arising from this submission. T.C. was supported by Austrian Science Fund Special Research Grant (FWF, SFB F79) and an ERC European Union's Horizon 2020 research and innovation program grant (AdG 694978). J.F.E. was supported by Austrian Science Fund DocFunds grant DOC 112-B. A.S. and V.B. are recipients of the DOC fellowship of the Austrian Academy of Sciences. Research at the IMP is supported by Boehringer Ingelheim and the Austrian Research Promotion Agency (Headquarter grant FFG-852936).

## Author contributions

Conceptualization, G.A.V., A.S., K.H., L.C. and T.C.; Methodology, G.A.V., A.S., K.H., J.F.E., V.B., A.M., D.B.G., J.B. and G.E.; Validation, A.S., K.H., J.F.E. and V.B; Formal Analysis, A.S. and K.H.; Investigation, A.S., K.H., J.F.E., V.B., A.M. and D.B.G.; Resources, J.F.E. and D.B.G.; Data Curation, A.S., K.H. and G.A.V.; Writing – Original Draft G.A.V. and A.S.; Writing – Review & Editing, A.S., K.H., J.F.E., V.B., A.M., D.B.G., L.C., T.C. and G.A.V.; Visualization A.S. and K.H.; Supervision G.A.V.; Project Administration G.A.V.; Funding Acquisition G.A.V.

## Competing interests

The authors declare no competing interests.
