## [Transparent Peer Review file · Nature Communications]

TRIM52 maintains cellular fitness and is under tight proteolytic control by multiple giant E3 ligases

Corresponding Author: Dr Gijs Versteeg

Version 0:

Reviewer comments:

Reviewer #1

(Remarks to the Author)

Previously, the Versteeg lab have demonstrated that TRIM52, a nuclear-localised ubiquitin E3 ligase, is one of the shortest lived proteins in the cell, with a half-life of ~3 minutes, which is attributable to the targeting of a long unstructured loop in its N-terminal RING domain (dubbed 'Loop 2', Hacker et al, Sci Rep, 2019). In this new manuscript, Shulkina et al. reveal that the rapid degradation of TRIM52 is brought about not by auto-ubiquitination, but instead by the sequential action of BIRC6 and UBR4/KCMF1. The authors find that whilst BIRC6 associates with TRIM52 via a largely charge-based association with 'Loop 2' to add the initial mono-ubiquitin, UBR4/KCMF1 binds elsewhere on the protein and then extends the ubiquitin chain. The data presented also suggest a supporting role for HUWE1, although the data and conceptual reasoning are not conclusive in this regard. Functionally, evidence is presented that suggests a role for TRIM52 in mediating transcription-induced DNA damage resolution, with DNA damage repair factors and transcriptional regulators enriched in the TRIM52 interactome. Additionally, etoposide-induced topoisomerase 2 association with DNA (indicating damaged DNA) was further enhanced when TRIM52 was knocked out, suggesting TRIM52 can mediate resolution of DNA damage stress.

These data add an interesting cross-regulatory aspect to our knowledge of ubiquitin E3 ligases. Whilst integrated, cross-talking networks are well-recognised for enzymatic cascades of other post-translational modifications (e.g. kinases), cross-regulation remains relatively underappreciated in the ubiquitin ligase field. A further point of interest in this paper is the co-operation of the large enzymatic structures of BIRC6 and UBR4/KCMF1 to drive polyubiquitination of TRIM52, which adds a further possible layer of regulation to TRIM52 ubiquitination.

The authors have employed a wide range of methodologies in their work, and the data presented are technically sound. The experiments presented are largely in accordance with the conclusions and claims the authors make. Nonetheless, I would like to offer the following comments and suggestions that I feel that would strengthen the manuscript.

General comments

- Whilst the title is memorable, it perhaps is too strongly worded. A) This paper does not have data regarding the 'primate-specific' expression of this ligase (this was found previously), B) Unless the data in Fig. 1 is strengthened, its relationship to DNA repair is unclear and not as conclusive as the title suggests, and C) The data regarding HUWE1 does not suggest it ubiquitinates TRIM52 as part of the same mechanism as BIRC6/UBR4. Therefore, the 'tight proteolytic control by a triad of giant E3 ligases' part of the title is not quite supported – the data for only two of the ligases clearly implicates them in TRIM52 ubiquitination and proteasomal degradation.

- TRIM52 is an unusual TRIM protein E3 ligase as it is missing the coiled coil and C-terminal domains generally found in TRIM family proteins. This should be briefly mentioned to provide context.

- The cropping of the following blots should be less stringent, with more of the surrounding membrane included: A) anti-GFP in Fig. 4g [particularly on the left hand side, where the band is cut in half]; B) anti-BIRC6 in Extended Data Fig. 5b [note, this is mislabelled in the source data as E.D. Fig. 5c]; and C) anti-BIRC6 for the input and IP in Extended Data Figure 5e [note, this is mislabelled in the source data as E.D. Fig. 5f].

- The diagrams of circles with lines in (presumably representing tagged protein) are not immediately interpretable by the reader (e.g Fig. 1d, Fig. 3b), as A) the circles do not look like plates of cells, and B) the protein inside each 'cell' is too small

to see when figures are printed.

- Although the authors demonstrate the localisation of TID-TRIM52 in Extended Figure 1e, they do not show the proper localisation of the other constructs (e.g. EGFP-TRIM52, mCherry-TRIM52). Given the possibility of fluorescent tags influencing localisation, it would be good to confirm their nuclear localisation.
- It is interesting to note that BIRC6, a key player in nuclear-localised TRIM52 ubiquitination, is not annotated as a nuclear-localised protein in the Human Protein Atlas or UniProt. The authors could confirm this by microscopy, and discuss possible mechanisms for BIRC6 nuclear translocation.
- What is the role of TRIM52 autoubiquitination and how does this relate to ubiquitination by BIRC6 and UBR4? Could the authors at least suggest a hypothesis?
- The authors mention cross-talk between TRIM52 and TRIM41. What is the evidence that this occurs on a functional level?
- It would assist the reader if the authors shared the data plotted in their proteomics volcano plots as Supplementary Excel files (in addition to their submission to the ProteomeXchange Consortium, as indicated in the Reporting Summary).
- The authors should clearly state what the error bars represent in each figure legend.

Figure 1

- Overall, the link with DNA damage is the least strong part of the manuscript, but has the potential to offer interesting findings of possible clinical significance for cancer treatments. Specifically, the authors suggest that TRIM52 is involved in DNA damage resolution. This would be more compelling if they could demonstrate by microscopy the formation of lesions (e.g. labelled by TOP2 or 53BP1) is increased in TRIM52 knockout (KO) cells or if they could demonstrate that lesions co-localise with TRIM52 after treatment with a DNA damaging agent, such as etoposide.
- These conclusions could be supported by a comet assay for DNA damage using WT vs TRIM52 KO cells.
- Although it might not flow in the current layout of the paper, it would be interesting to see whether the more stable TRIM52 Loop2 construct has any effect on TOP2 association with DNA after etoposide treatment. Additionally, it is important to test whether R187A or RING TRIM52 mutants influence TOP2 association with DNA (i.e. is it a ligase-dependent function?).

Figure 2

- It is intriguing that proteasome inhibition increases TRIM52 levels by only 2.5-fold in 4 h, given the described degradation time of ~3 mins as one would expect an increase of ~80-fold in 4 h ($(60 \times 4) / 3 = 80$)? Moreover, there doesn't appear to be a significant difference between 4 h and 8 h timepoints. Could the authors please comment on why they think this might be?
- Unfortunately, the authors are not able to reach a clear conclusion regarding which specific lysines are ubiquitinated, despite the use of appropriate methodologies, instead finding that they all have the potential to be ubiquitinated. However, they also state that in a lysine-less TRIM52 ubiquitination is reduced by only 70%. Could they please suggest whether they think other residues are being ubiquitinated to account for the remaining ~30% (e.g. the N-terminus)? Alternatively, could endogenous lysine-containing TRIM52 dimerise with the 'all K-to-R' mutant (possibly via the RING domain), thereby promoting its degradation and confounding this experiment?

Extended Data Figure 4

The authors state in the text that they 'targeted the individual E3 ligases in RKO cells stably expressing HA-tagged TRIM52 ...', but the figure does not include data for a HUWE1 knockout. Could the authors please clarify this in the text as the current wording is misleading given HUWE1's inclusion elsewhere in the manuscript.

Figure 4

- Could the authors please comment on the logic behind the choice of DAXX as a source of 'acidic amino acid composition', as it was not clearly introduced. Why this protein?
- To support the conclusion that the loop 2 region is a stand alone degron the authors should provide another example of protein degradation when fused to a neo protein.
- In Fig. 4f, lane 2 of the IP: a) There is actin contamination in the IP (indicating insufficient washing or contamination of the IP with the whole cell lysate), and b) a very weak co-IP with HUWE1. Therefore, HUWE1 may not be binding HA-TRIM52, but instead may be due to contamination.
- There is no loading control for Fig. 5g.

Figure 5

- Please could the authors specify in the figure legend which E1 and E2 enzymes were used for each reaction.
- Please could the authors clarify the identity of the upper band in Fig. 5a Coomassie stain?
- Similarly, could the authors please label the high MW band in Fig. 5d.
- Do the last 3 lanes on the right of Fig. 5f represent a timecourse? It is not immediately clear what the difference is between these samples.

(Remarks to the Author)

Reviewer #3

(Remarks to the Author)

In this study, the authors showed TRIM52's role in DNA damage response and its regulation through ubiquitination. TRIM52, identified as a critical player in repairing DNA damage, particularly involving the Homology-Dependent Repair (HDR) pathway, was found to interact genetically with the Non-Homologous End-Joining (NHEJ) pathway, suggesting a complementary role. Genetic screens revealed that TRIM52 ablation significantly reduces cellular fitness due to aberrant p53 activation and cell cycle arrest.

Through a series of genetic screens, researchers identified vital interactors, including TDP2 and ZNF451, essential for repairing double-strand breaks (DSBs) induced by topological stress during DNA replication and transcription. TRIM52 facilitates the repair of these lesions through the TDP2-dependent HDR pathway. In the RADAR assay, TRIM52-deficient cells showed increased covalent TOP2-DNA adducts and heightened sensitivity to Etoposide, highlighting TRIM52's role in resolving TOP2-associated DNA lesions.

TRIM52's degradation was primarily through the 26S proteasome, driven by K48-linked polyubiquitination. Genetic screens using a dual protein stability reporter system pinpointed specific E3 ligases, notably BIRC6, HUWE1, and UBR4/KCMF1, as regulators of TRIM52 degradation. Validation experiments confirmed these enzymes' roles, showing that knockout of these E3 ligases increased TRIM52 levels and stability.

The study further elucidated that TRIM52's rapid degradation is attributed to its RING domain's loop two regions, a crucial factor. Mutations in loop two or its deletion significantly stabilised TRIM52, indicating its importance in TRIM52 ubiquitination and recognition by E3 ligases.

In vitro assays demonstrated that TRIM52 auto-ubiquitinates and BIRC6 efficiently mono-ubiquitinates TRIM52 on specific residues. UBR4/KCMF1 and HUWE1 did not show robust activity alone but could extend BIRC6-seeded mono-ubiquitination into poly-ubiquitin chains. This cooperative mechanism suggests that BIRC6 initiates TRIM52 mono-ubiquitination on its RING domain, which UBR4/KCMF1 subsequently extends into poly-ubiquitin chains, crucial for TRIM52's targeted degradation.

Overall, these findings advance our understanding of TRIM52's role in DNA damage repair and its regulation via ubiquitination, highlighting the coordinated efforts of multiple E3 ligases in modulating its stability and function. However, the study needs significant improvement.

Major Comments:

1. The major finding regarding the repair of TDP2 lesions via the Homology-Dependent Repair (HDR) pathway requires additional support. While the study shows the Non-Homologous End-Joining (NHEJ) pathway activation following TRIM52 ablation, it would be beneficial to conduct similar experiments depleting RAD51 or BARD1, both critical for the HDR pathway. As these genes are highly enriched in one of the genetic screens (Figure 1h), their depletion would provide more precise insights into HDR involvement.
2. The abstract claims, "Genetic and proteomic analyses revealed TRIM52's involvement in resolving topoisomerase 2 (TOP2)-DNA cross-links, mitigating DNA damage, and preventing cell-cycle arrest." We suggest that additional experiments be performed to elucidate the molecular pathways of TDP2-dependent repair in a TRIM52-dependent manner, providing robust evidence for these claims.
3. In Figure 3b, the blot images for TRIM52 appear faint, possibly due to TRIM52's short half-life, and the actin bands are also faint. We encourage the authors to provide clarification and, if possible, improve the quality of these blot images to ensure accurate interpretation.
4. Figure 4b claims that only partial ubiquitination exists in the loop of two neutral mutants, but the smear appears very similar to the full-length protein. A more apparent distinction or further clarification would strengthen this point.
5. In Figure 4f, the authors state that HUWE1 binding is reduced when loop two is depleted; however, the bands for HUWE1 and BIRC6 are not prominent. More explicit immunoprecipitation images are recommended to substantiate this observation.
6. For extended Figure 3b, it would be helpful to see the effects of epoxomicin treatment at various time points, similar to those shown for cycloheximide (CHX), to better understand its impact over time.
7. The authors are encouraged to show the effect of sgHUWE1 on TRIM52 turnover in a CHX chase experiment. Additionally, the band intensity difference in the sgUBR4 blot is noticeable compared to sgKCMF1 and sgBIRC6. Clarification is needed on whether UBR4 knockout affects TRIM52 morphology or proliferation.
8. It would be beneficial to examine the autoubiquitination effect of the R187A mutant in an in-vivo model and assess the stability of TRIM52 with this mutation.

Minor comments:

1. In Figure 2i, displaying the entire band of 3xHA TRIM52 would be beneficial to confirm the absence of ubiquitination in the KtoR mutant following Etoposide treatment.
2. There are a few spelling errors that need correction: In Figure 2f, "ubiquitin" is misspelled, in Figure 2h, "identified" is misspelled in the second bullet point, and in the Figure 4 legends (g), "form" should be corrected to "from."
3. The abbreviations for Etoposide and Epoxomicin are unclear. It would be helpful to define these abbreviations in the main text or figure legends.
4. For Figure 3g, please include larger images of the TRIM52 blot. Additionally, the font size in the labels on the right side needs to be increased, and the text "endogenous TRIM52" should be corrected.

5. In Figure 5a, clarification is needed on the difference between lanes 4 and 5, as the dots indicate the same genes and enzymes.
6. Finally, in Figure 5f, there is a discrepancy in the GFP blot compared to Figure 5g. A higher exposure image of the blot would be appreciated to provide more precise results.

Reviewer #4

(Remarks to the Author)

TRIM52 is a primate-specific ubiquitin ligase with little known function. Shulkina and colleagues seek to uncover the biological function of TRIM52 using a synthetic lethal CRISPR screen. The authors show that the loss of TRIM52 is critical for cell proliferation and synergizes with a group of DDR (DNA damage response) genes. Additionally, the authors find that TRIM52 may physically interact with an entirely different set of DDR factors using an unbiased proximity labeling-based approach. The authors then come up with a model for how TRIM52 might play a role in mitigating DNA damage by resolving TOP2-DNA crosslinks. Unfortunately, relatively little attention is given to how TRIM52 fits into the DDR pathway. Instead, the authors spend most of their time describing how the stability of TRIM52 is regulated by a group of giant protein quality control E3 ligases under normal growth conditions (and not under DNA damage). Though their characterization of TRIM52 stability by these E3s is nicely done, the purpose of this characterization remains unclear given the tenuous connection between TRIM52 and the DDR.

Major points

1. The DEPMap database shows that TRIM52 is non-essential in RKO cells. Though I am happy to concede that DEPMap may be wrong, the authors need to rescue their Δ TRIM52 cell proliferation defect by reintroducing exogenous TRIM52.
2. From their synthetic lethal CRISPR screen, the authors find many DDR genes that synergize with the loss of TRIM52. What's perplexing is that these DDR genes aren't really interrelated but a smattering of DDR factors that work to repair distinct types of DNA damage (e.g. replication stress-induced breaks, mismatches, double-stranded breaks). Additionally, while NHEJ1 is their top hit, very few of the other canonical members of the NHEJ pathway, outside of XRCC4, came up as hits (e.g. KU70, KU80, etc). While it's possible that these genes, being essential, might have dropped out of the screen, the authors should perform competition assays to validate whether the loss of other NHEJ genes also compromises cell fitness in the context of TRIM52 loss. Further, there is very little overlap between this gene set and their unbiased proximity-labeled screen (Fig. 1h), complicating the biological relevance of their hits.
3. The biggest flaw with this manuscript is that the authors use a defect in cell proliferation as a proxy for DNA damage (Fig. 1g). Though the authors show that loss of TRIM52 synergizes with DDR genes, this synergy could have nothing to do with DNA damage. The authors need to show that loss of TRIM52 results in DNA damage (e.g. alkaline comet assays, gammaH2AX staining). What's more puzzling is that the author's previous publication (Benke et al. 2018) shows that loss of TRIM52 causes cells to exit from the cell cycle and remain in a G0/G1 state. Thus, their observed proliferation defect could simply be due to this cell cycle exit. This huge unknown makes evaluating the manuscript premature. For example, if loss of TRIM52 causes DNA damage, what types of breaks does it make? Which substrates of the DDR signaling pathway are regulated by TRIM52? Is the ligase activity of TRIM52 required during the DDR or does TRIM52 have ligase-independent functions? If loss of TRIM52 does not cause DNA damage, on the other hand, what is the biological relevance of the observed synergy between the TRIM52 and DDR genes?
4. The authors should show their studies using overexpressed TRIM52 (TurboID, puncta formation in the nucleus, etc) is not an overexpression artifact. Though they sought to match the expression of TRIM52 to GFP, they should have matched it to endogenous TRIM52. Or at least validate their proximity labeling studies in some way.
5. The authors nicely show, by using their cellular fluorescent reporter system, that loss of HUWE1, BIRC6, and UBR4 causes stabilization of the GFP-TRIM52 reporter. Yet, their in vitro reconstitution of TRIM52 ubiquitination remains unconvincing (Figs. 5e and f). Figures 5e and f show that HUWE1, BIRC6, and UBR4 are active ubiquitin ligases and BIRC6 likely monoubiquitates TRIM52. However, none of the ligases seem to polyubiquitate TRIM52 in their purified system, given there is a complete lack of higher-order molecular weight species or a concomitant loss of the unmodified form in their aGFP blots. The ubiquitin chains do not seem to be covalently linked to TRIM52 but co-IP indirectly.

Minor points

1. Why are there no polyubiquitin chains in the mCherry IP for the MYC-mCherry-Trim52 construct for Figure 2g when these chains are clearly visible in Fig. 5e?
2. Which lysines were mutated in the 5KtoR construct? Did the authors identify the incorrect lysines (Figure 2l)?

Reviewer #5

(Remarks to the Author)

I co-reviewed this manuscript with one of the reviewers who provided the listed reports. This is part of the Nature

Communications initiative to facilitate training in peer review and to provide appropriate recognition for Early Career Researchers who co-review manuscripts.

Version 1:

Reviewer comments:

Reviewer #1

(Remarks to the Author)

The authors have satisfactorily addressed all concerns raised. The manuscript is now suitable for publication.

Reviewer #2

(Remarks to the Author)

Reviewer #3

(Remarks to the Author)

The authors have significantly improved the revised version with the additional set of experiments, clarified some aspects of their findings in the text, and addressed some of my concerns about their first version.

However, the revised version fails to convince me that TRIM52 is involved in Top2-cc repair. If the authors want to emphasize that TRIM52 indeed works as one of the pathways for Top2cc repair, they have to provide more solid data. In addition, Fig 2f is of very poor resolution, and nuclei are much larger in sgTRIM52/ETO treated cells, than in sgAAVS1/ETO treated cells; so the number of increased γ -H2AX foci/nucleus could be significant as the nuclei are just simply larger in sgTRIM52/ETO than in sgAAVS1/ETO.

Regardless, the author should perform a better set of experiments to convince me that TRIM52 is indeed involved in Top2-cc repair.

For instance:

- 1) The authors must perform RADAR assay with ETO treatment (to induce a high level of TOP2cc) and then remove the drug and monitor TOP2cc repair during the recovery time (e.g. 1 hr, 2 hr, 4 hr and 8 hr). This would be the 1st body of evidence that TRIM52 plays a role in TOP2cc resolution.
- 2) The cell cycle analysis (FACS) must support these experiments to show the effect of TRIM25-ko on the cell cycle under unperturbed conditions, in response to ETO and during the recovery time.
- 3) These FACS data should be accompanied by a γ -H2AX signal to demonstrate the level of DSB activation. Alternatively, the authors can perform a Comet assay to show that DSBs persist or their repair is delayed in TRIM25-ko cells (during the recovery time), and these breaks depend on TOP2cc (after ETO treatment).
- 4) They also have to perform a CFA assay and confirm that TRIM52 cells are hypersensitive to ETO, and this hypersensitivity is a) abolished when TOP2 is removed (e.g. si or shTOP2) and b) restored when TRIM52wt is expressed back.
- 5) If TOP2cc indeed persists in TRIM52 cells, they should also be able to visualize the persistence of DSBs by IF showing a γ -H2AX/53BP1 colocalization signal.
- 6) Throughout these experiments, a positive control for abolished TOP2cc repair (e.g., siZNF45 or Rad54L2) must also be included.
- 7) I suggest they check this recent paper on Top2cc repair: <https://www.science.org/doi/10.1126/sciadv.adl2108> and use it as an example how to demonstrate/confirm the potential role of TRIM52 in Top2cc repair.

Reviewer #4

(Remarks to the Author)

Shulkina and colleagues seek to decipher the functional relevance of TRIM52 in DNA repair biology and better understand the regulation underlying TRIM52 stability. The authors claim that TRIM52 is involved in DNA repair because TRIM52 “synergizes” with multiple DNA repair factors in their CRISPR screen. However, the biological role of TRIM52 and its mechanism of action in DNA repair remain unresolved.

1. Generally speaking, the loss of essential DNA repair factors causes cell death by inducing genomic instability in the form of irreversible DNA breaks, and the upregulation of γ -H2AX is often used as an indicator of DNA damage and cell death if the damage is not resolved in a timely manner. The authors show that TRIM52 is essential (Fig. 1g), yet its loss does not increase DNA damage (Figs. 1d-g) without the addition of an external source of DNA damage. This indicates to me that the essentiality of TRIM52 is not due to its role in DNA repair but through another function (perhaps DNA replication). This issue makes their claims about epistatic relationships in the context of DNA repair unclear (lines 104-110).

2. I remain unconvinced by the author's in vitro reconstitution experiments (Fig. 6F). The authors claim that BIRC6 and

UBR4 work together to polyubiquitinate GFP-TRIM52, yet I do not see any higher molecular weight species in their anti-GFP blots (either in their inputs or IPs). Curiously, the authors instead blot for ubiquitin (anti-FLAG) and find an increase in ubiquitin signal only when they IP their substrate under native IP conditions. This simply tells me that ubiquitin chains co-associate with GFP-TRIM52 indirectly instead of being covalently linked to GFP-TRIM52 (because there are no higher molecular weight species in the corresponding anti-GFP blots). If the authors want to show that BIRC6/HUWE1 are covalently modifying GFP-TRIM52 with ubiquitin chains using this type of assay, they really need to perform their IPs under denaturing (or non-native) conditions.

Reviewer #5

(Remarks to the Author)

Rebuttal letter for Nature Communications manuscript NCOMMS-24-30996-T

"TRIM52 maintains genomic DNA integrity and is under tight proteolytic control by multiple giant E3 ligases" Alexandra Shulkina *et al.*

We would like to very much thank the reviewers for their efforts and constructive feedback, which helped to substantially improve our revised manuscript. Please find below an overview of the changes in the revised manuscript, followed by a point-by-point rebuttal.

Overview of changes:

In total 15 new data figure panels were generated, and included in the revised manuscript

1. Due to the volume of newly added data, Fig.1 has been split into two figures: Fig.1 and Fig.2.
2. New data, added as Figure 1 – **Extended Fig.1b**. These data address how the KO of TRIM52 in combination with NHEJ and HDR factors affects cellular fitness.
3. New data, added as Figure 2 – **Fig.2f-g**. These data address induction of DNA damage upon *TRIM52* ablation.
4. New data, added as Figure 2 – Extended data **Fig.2 e-f**. These data address the localization of TRIM52 fused to different fluorophores EGFP and mCherry upon proteasomal inhibition.
5. New data, added as Figure 2 – Extended data **Fig.2g**. These data address the localization of endogenous TRIM52 upon proteasomal inhibition with epoxomicin.
6. New data, added as Figure 4 – Extended data **Fig.4b-c**. These data address the cellular kinetics of TRIM52 protein level increase upon proteasomal inhibition over time.
7. New data, added as Figure 4 – Extended data **Fig.5c-d**. These data address how half-life of TRIM52 increases upon ablation of the identified E3 ligases identified in the genetic screen.
8. New data, added as Figure 5 – **Figure 5f**. These data show that wild-type TRIM52 reproducibly co-IPs with HUWE1.
9. New data, added as Figure 5 – **Extended data Fig.6b**. These data address Loop2 degron quality and ability to destabilize the otherwise stable EGFP and mCherry proteins.
10. New data, added as Figure 5 – **Extended data Fig.6f**. These data address the contribution of auto-ubiquitination to TRIM52 degradation. We show that catalytically inactive TRIM52-R187A is equally unstable as its wild-type counterpart.
11. New data, added as Figure 5 – **Extended data Fig.6i**. These data address how TRIM52-WT and TRIM52 (Δ Loop2) co-IP with the identified E3 ligases from the genetic screen.
12. New data, added as Figure 6 – **Extended data Fig.7f**. These data address cellular localization of endogenous BIRC6.

Point-by-point rebuttal

Reviewer comments are in *blue cursive*.

Reviewer #1:

1. *Whilst the title is memorable, it perhaps is too strongly worded. A) This paper does not have data regarding the 'primate-specific' expression of this ligase (this was found previously), B) Unless the data in Fig. 1 is strengthened, its relationship to DNA repair is unclear and not as conclusive as the title suggests, and C) The data regarding HUWE1 does not suggest it ubiquitinates TRIM52 as part of the same mechanism as BIRC6/UBR4. Therefore, the 'tight proteolytic control by a triad of giant E3 ligases' part of the title is not quite supported – the data for only two of the ligases clearly implicates them in TRIM52 ubiquitination and proteasomal degradation.*

Reply: We agree with points A and C. Moreover, we have included additional experimental data strengthening a role of TRIM52 in preventing TOP2-dependent DNA damage (Fig. 2f-g). Based on this, the title of the manuscript was changed to “TRIM52 maintains genomic DNA integrity and is under tight proteolytic control by multiple giant E3 ligases”.

2. *TRIM52 is an unusual TRIM protein E3 ligase as it is missing the coiled coil and C-terminal domains generally found in TRIM family proteins. This should be briefly mentioned to provide context.*

Reply: We agree. Text providing this context was added to the manuscript introduction section (lines 53-56).

3. *The cropping of the following blots should be less stringent, with more of the surrounding membrane included: A) anti-GFP in Fig. 4g [particularly on the left hand side, where the*

band is cut in half]; B) anti-BIRC6 in Extended Data Fig. 5b [note, this is mislabelled in the source data as E.D. Fig. 5c]; and C) anti-BIRC6 for the input and IP in Extended Data Figure 5e [note, this is mislabelled in the source data as E.D. Fig. 5f].

Reply: We agree and adjusted the cropping accordingly. The adjusted data figures are included in the revised manuscript as Fig. 5h, Extended data figure 6c, and Extended data figure 6g, respectively. The mislabeling for the latter two panels has been corrected as well.

h

c

g

4. The diagrams of circles with lines in (presumably representing tagged protein) are not immediately interpretable by the reader (e.g Fig. 1d, Fig. 3b), as A) the circles do not look like plates of cells, and B) the protein inside each 'cell' is too small to see when figures are printed.

Reply: The figure sizes were increased, and the figure legends for both panels were adjusted to clarify that the circles represent individual cells, and the entities inside the cells differentially color-coded sgRNAs.

d

b

5. Although the authors demonstrate the localisation of TID-TRIM52 in Extended Figure 1e, they do not show the proper localisation of the other constructs (e.g. EGFP-TRIM52, mCherry-TRIM52). Given the possibility of fluorescent tags influencing localisation, it would be good to confirm their nuclear localisation.

Reply: New data addressing the localization of used constructs and endogenous TRIM52 have been included. Specifically, the updated version of Extended Fig.2d-g now includes immunofluorescence microscopy assay data for EGFP-TRIM52 and mCherry-TRIM52. These results show that both of these constructs localize to the nucleus upon inhibition of proteasomal degradation with epoxomicin, similarly to TurboID-TRIM52. Moreover, additional subcellular fractionation experiments showed that endogenous TRIM52 accumulates in the nucleus in response to epoxomicin treatment. Together, these data indicate that TurboID-, or fluorophore-tagged TRIM52 variants localize in a manner comparable to endogenous TRIM52, and that their unstable sub-cellular fraction is most likely nuclear.

6. *It is interesting to note that BIRC6, a key player in nuclear-localised TRIM52 ubiquitination, is not annotated as a nuclear-localised protein in the Human Protein Atlas or UniProt. The authors could confirm this by microscopy and discuss possible mechanisms for BIRC6 nuclear translocation.*

Reply: To address this, RKO cells were fractionated, and BIRC6 analyzed in the cytosolic and nuclear fractions. While we found BIRC6 to be predominantly present in the cytoplasm, we also detected a small fraction in the nucleus (Extended data Fig. 7f). This suggests that there may be some nuclear BIRC6 that could directly mono-ubiquitinate TRIM52 in the nucleus, which subsequently facilitates its poly-ubiquitination and degradation. However, we cannot exclude the alternative option that TRIM52 is mono-ubiquitinated by BIRC6 in the cytoplasm, yet poly-ubiquitinated and degraded after translocation to the nucleus. Text was added to the Results section to convey this notion (lines 440-448).

7. *What is the role of TRIM52 autoubiquitination and how does this relate to ubiquitination by BIRC6 and UBR4? Could the authors at least suggest a hypothesis?*

Reply: We previously reported that a cysteine mutant in the TRIM52 RING domain was still highly unstable, comparable to its wild-type counter-part (Hacker *et.al*, Scientific Reports, 2018). As indicated in the manuscript text, this has suggested that its own E3 ligase activity is dispensable for TRIM52 turn-over. To further substantiate this finding with a subtler mutagenesis approach, we tested the genetic interaction of the TRIM52-R187A linchpin mutant (which lacks E3 ligase activity *in vitro*) in *UBR4*, *HUWE1*, or *BIRC6* knock-out cells.

Additional cycloheximide chase experiments of catalytically inactive TRIM52-R187A show that its half-life is still very short (31.6 min.) and it is turned-over in a proteasomal-dependent manner (Extended data Fig.6f). This is comparable to exogenously expressed catalytically active TRIM52, which had a measured half-life of 36.2 min.

8. *The authors mention crosstalk between TRIM52 and TRIM41. What is the evidence that this occurs on a functional level?*

Reply: We briefly discuss the evolutionary connection to TRIM41 in the discussion section, and point out that possibly both proteins could be functionally connected as they both regulate different topoisomerases. However, this text was only included to point out this speculative possibility. We neither have any experimental evidence for such a connection, nor claim that a functional connection with TRIM41 exists. The relevant section in the discussion text has been changed to reflect this notion (Lines: 473-474).

9. *It would assist the reader if the authors shared the data plotted in their proteomics volcano plots as Supplementary Excel files (in addition to their submission to the ProteomeXchange Consortium, as indicated in the Reporting Summary).*

Reply: These data were already included in a table format in the original manuscript submission, and are still part of this resubmission (Supplementary data 2).

10. The authors should clearly state what the error bars represent in each figure legend.

Reply: Error bars reflect standard deviations. This important information was indeed missing from the legend for some of the figure panels. Figure legends for all relevant figure panels have been updated to include this information.

Reviewer #2:

1. Specifically, the authors suggest that *TRIM52* is involved in DNA damage resolution. This would be more compelling if they could demonstrate by microscopy the formation of lesions (e.g. labelled by TOP2 or 53BP1) is increased in *TRIM52* knockout (KO) cells or if they could demonstrate that lesions co-localise with *TRIM52* after treatment with a DNA damaging agent, such as etoposide.

Reply: To substantiate whether *TRIM52* loss results in additional marks of DNA damage, sgAAVS1- and sg*TRIM52*-targeted RKO-Cas9 cells were treated with or without etoposide, and subsequently analyzed by γ H2AX staining. *TRIM52* ablation increased the number of γ H2AX foci per nuclei compared to sgAAVS1-targeted control cells. As we reported for TOP2 adducts, loss of *TRIM52* in untreated cells already showed a trend for increased γ H2AX foci in *TRIM52* KO cells, whereas in the presence of etoposide this difference was stronger and statistically significantly different between the two genotypes. This result is consistent with our observation that loss of *TRIM52* diminishes the resolution of covalently associated TOP2 with the DNA, resulting in formation of double-strand breaks. These new data were added to the revised manuscript as Fig.2 f-g.

2. *Although it might not flow in the current layout of the paper, it would be interesting to see whether the more stable TRIM52 Δ Loop2 construct has any effect on TOP2 association with DNA after etoposide treatment. Additionally, it is important to test whether R187A or Δ RING TRIM52 mutants influence TOP2 association with DNA (i.e. is it a ligase-dependent function?).*

Reply: We agree that it is an interesting biological question whether TRIM52 turn-over has a functional role in TRIM52 function, which we intend to further address outside the scope of this manuscript. This mainly stems from the fact that this would likely take a significant additional time investment for setting up a functional rescue system to address functionality of TRIM52 mutants.

Prior to the initial submission and during the revision process we extensively invested in various approaches to functionally rescue *TRIM52* knock-out (sgRNA) and knock-down (shRNA) cells with wild-type and mutant sgRNA/shRNA-resistant TRIM52 cDNA expression constructs. We attempted expression of tagged and untagged TRIM52 versions, different promoter strengths for driving transgene expression of all reported TRIM52 isoforms. Unfortunately, none of these tested conditions functionally rescued fitness of *TRIM52* loss-of-function cells in competition experiments.

Although we do not have definitively determined why these rescue experiments did not work, we speculate that substantially higher transgene expression than endogenous TRIM52 (more than 10-fold even with the weakest tested promoter) could hinder functional rescue. We plan to address this question in the future by engineering of the endogenous *TRIM52* locus in order to maintain correct expression levels.

We want to emphasize that our results show that our functional TRIM52 data are specific (yet may require endogenous TRIM52 levels), whereas TRIM52 protein degradation phenocopies the endogenous protein independent of expression levels and tags. Specifically, we have targeted *TRIM52* with three independent shRNAs (Benke *et al.* Oncotarget 2018), and seven independent sgRNAs (Hacker *et al.* Scientific Reports 2019, and current manuscript), which all reduced cell fitness. Moreover, the magnitude of fitness-loss scaled with the strength of TRIM52 protein depletion. This strongly indicates that the measured loss-of-TRIM52 effects are specific and not the results of off-target effects.

3. *It is intriguing that proteasome inhibition increases TRIM52 levels by only 2.5-fold in 4 h, given the described degradation time of ~3 mins as one would expect an increase of ~80-fold in 4 h ((60x4)/3=80)? Moreover, there doesn't appear to be a significant difference between 4 h and 8 h timepoints. Could the authors please comment on why they think this might be?*

Reply: This is a more general phenomenon that we have seen for all unstable proteins that we have investigated. We think that this is determined by two factors: 1) transcriptional strength, and 2) increase relative to the pre-existing steady-state protein pool.

Our interpretation of this phenomenon has been that in the presence of complete proteasomal degradation block, steady-state levels are exclusively affected by *de novo* synthesis. We speculate that translation up to 6 h during epoxomicin treatment is fairly constant, and that this practically means that in most cases transcription rates determine how much protein levels will be increased during the time that protein degradation is prevented.

We have previously identified a rather weak promoter driving low constitutive expression of endogenous TRIM52, whereas the SFFV promoter for exogenous expression is considered a stronger 'medium-strength' promoter. We believe that this importantly contributes to the modest increases in protein steady-state levels during proteasome-block.

In addition, since synthesis rates for a given protein are mostly constant for a given promoter, the relative fold change in protein steady-state levels is affected by the pre-existing protein pool size established over a much longer time than the subsequent time-frame of inhibitor treatment. We think that proteins expressed from stronger promoters (such as the SFFV) have higher protein pools prior to proteasome-block, which may limit the fold-increase relative to this starting protein pool size during proteasome inhibitor treatment despite having stronger synthesis during the relatively short six-hour window.

Lastly, based on cell morphology and previous viability measurements, we think that after 6 h of proteasome inhibition inhibitor-toxicity likely decreases synthesis rates, which in turn diminish additional accumulation of stabilized protein.

- 4. Unfortunately, the authors are not able to reach a clear conclusion regarding which specific lysines are ubiquitinated, despite the use of appropriate methodologies, instead finding that they all have the potential to be ubiquitinated. However, they also state that in a lysine-less TRIM52 ubiquitination is reduced by only 70%. Could they please suggest whether they think other residues are being ubiquitinated to account for the remaining ~30% (e.g. the N-terminus)? Alternatively, could endogenous lysine-containing TRIM52 dimerise with the 'all K-to-R' mutant (possibly via the RING domain), thereby promoting its degradation and confounding this experiment?*

Reply: Based on our mass-spectrometry data, we concluded that there are likely two predominant lysines that are ubiquitinated in TRIM52. However, as is the case for many unstable proteins, mutation of these lysines allows compensatory targeting of other lysines given spatial flexibility of a large fraction of E3 ligases.

We acknowledge that even in the absence of lysines the protein is not completely stable, and agree that these are two feasible possibilities as to why we did not measure complete stabilization of the lysine-less variant. It could be that non-lysine ubiquitination through oxy-esters on non-lysine residues plays a minor role. In this context, the E3 ligase MYCBP2 which has been reported to catalyze such ubiquitin conjugates was a medium-strength hit in our screen. However, we did not functionally test the role of non-lysine ubiquitination, or the second option (dimer formation with endogenous TRIM52), or N-terminal ub-chain conjugation experimentally. We have added text to the Results section discussing these possibilities (lines: 207-209).

- 5. The authors state in the text that they 'targeted the individual E3 ligases in RKO cells stably expressing HA-tagged TRIM52 ...', but the figure does not include data for a HUWE1 knockout. Could the authors please clarify this in the text as the current wording is misleading given HUWE1's inclusion elsewhere in the manuscript.*

Reply: The CHX chase experiment was repeated with a complete set of E3 ligase KOs, including HUWE1 (Extended data Fig.5c-d). Consistent with our other findings, the results demonstrated that ablation of all of the identified E3 ligases in our screen increased the half-life of TRIM52.

c**d**
6. *Could the authors please comment on the logic behind the choice of DAXX as a source of 'acidic amino acid composition', as it was not clearly introduced. Why this protein?*

Reply: DAXX is to the best of our knowledge one of the few proteins which has a similarly biased protein region comparable to the TRIM52 RING loop 2 in terms of acidic amino acid composition and length. As such, we reasoned it to be a good comparative control to determine how biased, acidic composition *per se* influences protein stability in comparison to TRIM52-loop2-specific features within a region with such a composition. We have added additional text in the revised manuscript in the Results section to clarify this point (lines: 312-317).

7. *To support the conclusion that the loop 2 region is a standalone degron the authors should provide another example of protein degradation when fused to a neo protein.*

Reply: To test whether the loop2 region would destabilize another protein, it was fused to the stable EGFP protein. These data were added to the revised manuscript as Extended data Fig.6b. The fusion of the TRIM52-loop2 region to EGFP rendered it unstable in a fashion similar to TRIM52-loop 2 fusion to mCherry. The results showed that while EGFP alone did not increase upon treatment of cells with epoxomicin, EGFP-Loop2 significantly increased by approximately 2-fold.

b
8. *In Fig. 4f, lane 2 of the IP: a) There is actin contamination in the IP (indicating insufficient washing or contamination of the IP with the whole cell lysate), and b) a very weak co-IP with HUWE1. Therefore, HUWE1 may not be binding HA-TRIM52, but instead may be due to contamination.*

Reply: Additional co-IP experiments were performed without contamination to address this point, the data of which were added as new Fig. 5f. From these new results we conclude that wild-type TRIM52 reproducibly co-IPs with HUWE1 in the absence of contamination, and thus that the minor Actin contamination in Fig.5g does not affect the overall conclusion regarding how deletion of the Loop2 region affects binding by the E3s. We also included additional co-IP data within the revised manuscript as Extended data Fig.6i to showcase the reproducibility of the TRIM52 interaction with HUWE1 and BIRC6 in cells.

9. *There is no loading control for Fig. 5g.*

Reply: This is a co-IP with purified recombinant proteins, there is no loading control.

10. *Please could the authors specify in the figure legend which E1 and E2 enzymes were used for each reaction.*

Reply: Different samples contained different enzyme combinations. We tried adding the different E1/E2 enzyme information in the figures and/or the legends, but in our opinion, this decreased clarity. Therefore, we included all information in tabular format. Supplementary data 6 now describes all reaction components of the *in vitro* ubiquitination reactions in Fig.6 and Extended data Fig. 7. Text to make this clear to readers was added in the revised manuscript.

11. *Please could the authors clarify the identity of the upper band in Fig. 5a Coomassie stain?*

Reply: That band is the E1 enzyme, and is now labeled in the updated figure panel.

12. Similarly, could the authors please label the high MW band in Fig. 5d.

Reply: That band is BIRC6, and is now labeled in the updated figure panel.

13. Do the last 3 lanes on the right of Fig. 5f represent a time course? It is not immediately clear what the difference is between these samples.

Reply: The last three lanes represent replicates of the same reaction in different buffer conditions. This is explained in the figure legend; additional text to further clarify this was added to the results section of the revised manuscript (lines: 436-437).

Reviewer #3:

Major Comments:

1. *The major finding regarding the repair of TDP2 lesions via the Homology-Dependent Repair (HDR) pathway requires additional support. While the study shows the Non-Homologous End-Joining (NHEJ) pathway activation following TRIM52 ablation, it would be beneficial to conduct similar experiments depleting RAD51 or BARD1, both critical for the HDR pathway. As these genes are highly enriched in one of the genetic screens (Figure 1h), their depletion would provide more precise insights into HDR involvement.*

Reply: Additional competition assays were performed with cells in which individual HDR (*RAD51* and *BARD1*) or NHEJ (*Ku70* and *Ku80*) components were targeted, in combination with *TRIM52* ablation. These new data were added as Extended Data Figure 1a-b to the revised manuscript.

In contrast to our previous speculation that *TRIM52* could function upstream of the HDR pathway, loss of both NHEJ and HDR components had additive loss-of-fitness effects to *TRIM52* ablation. These non-epistatic interactions with both the NHEJ and HDR pathways suggest that *TRIM52* can redundantly function in parallel to either repair pathway, e.g. in a partially functionally manner. Both HDR and NHEJ have been reported to repair TOP2-induced lesions, based on these findings we therefore hypothesize that *TRIM52* acts in parallel to both of these pathways, and that in the absence of *TRIM52* cells rely significantly more on these DNA repair pathways to resolve TOP2-mediated damage. The manuscript text has been adjusted to reflect our new findings and interpretation of the data.

The fact that core components of neither repair pathway were identified as hits in the suppressor screen is consistent with the pre-dropout screening strategy, which removed cells targeting essential genes.

2. *The abstract claims, "Genetic and proteomic analyses revealed TRIM52's involvement in resolving topoisomerase 2 (TOP2)-DNA cross-links, mitigating DNA damage, and preventing cell-cycle arrest." We suggest that additional experiments be performed to elucidate the molecular pathways of TDP2-dependent repair in a TRIM52-dependent manner, providing robust evidence for these claims.*

Reply: Additional DNA damage experiments were performed to further analyze the role of TRIM52 in restricting DNA damage. Data from these experiments were added as new Fig.2 f-g. To this end, sgAAVS1 or sgTRIM52 cells treated with or without the TOP2 poison Etoposide were analyzed by γ H2AX staining. Upon ablation of TRIM52, we measured a trend (albeit non-significant) for an increased number of γ H2AX foci in TRIM52 KO cells in DMSO-treated control cells. Importantly, in the presence of Etoposide, loss of TRIM52 significantly increased the number of γ H2AX-marked DNA lesions. This result is consistent with our RADAR assay data that loss of TRIM52 leads to a defect in the resolution of covalently associated TOP2 from the DNA, resulting in the formation of double strand breaks.

3. *In Figure 3b, the blot images for TRIM52 appear faint, possibly due to TRIM52's short half-life, and the actin bands are also faint. We encourage the authors to provide clarification and, if possible, improve the quality of these blot images to ensure accurate interpretation.*

Reply: Since Fig. 3b is not a Western blot image, we assumed the reviewer meant Fig. 1b. In Fig. 1b, the endogenous TRIM52 band is indeed faint because of the exceptionally low steady-state protein levels resulting from its short half-life. Low actin exposures were included in the original figure panel to ensure that it is visually clear to readers that even though more total lysate was loaded in the knock-out sample, the targeted TRIM52 protein is undetectable. We replaced the actin panel with a higher exposure image in the revised manuscript. In the current state we think that it improves the figure, while still accurately representing the increased loading in the knock-out sample. Importantly, despite more loading in the knock-out sample, TRIM52 is undetectable.

b
4. *Figure 4b claims that only partial ubiquitination exists in the loop of two neutral mutants, but the smear appears very similar to the full-length protein. A more apparent distinction or further clarification would strengthen this point.*

Reply: We agree and also concluded that the ubiquitination of the loop2-neutral mutant is comparable to wild-type TRIM52. The manuscript text was adapted to clarify and reflect this conclusion (lines: 325-327). This is consistent with the observation that this mutant is also still unstable and increases in the presence of epoxomicin.

5. *In Figure 4f, the authors state that HUWE1 binding is reduced when loop two is depleted; however, the bands for HUWE1 and BIRC6 are not prominent. More explicit immunoprecipitation images are recommended to substantiate this observation.*

Reply: Detection of the studied large E3 ligases by WB is technically challenging as a result of their cumbersome Western transfer, and lack of exceptional antibodies for detection. With our current (optimized) systems it has not been technically possible to further improve the quality of the co-IP at this moment. However, we included additional co-IP HUWE1 data in the revised manuscript as Fig.5f to showcase the reproducibility of the TRIM52-HUWE1 interaction in cells. We also included additional co-IP data within the revised manuscript as Extended data Fig.6i to showcase the reproducibility of the TRIM52 interaction with HUWE1 and BIRC6 in cells. We would like to emphasize that although the HUWE1 and BIRC6 bands are rather low-intensity in all detections, we have shown specificity of the signal in *HUWE1*- and *BIRC6*-targeted cells, and reproducibility of specific co-IP using no-bait controls.

f**i**
6. For extended Figure 3b, it would be helpful to see the effects of epoxomicin treatment at various time points, similar to those shown for cycloheximide (CHX), to better understand its impact over time.

Reply: A time-course during epoxomicin treatment was performed, and TRIM52 protein levels analyzed. These new data were included in the revised manuscript as Extended data Fig. 4b-c. Consistent with its proteasomal degradation, TRIM52 protein levels increased over time.

7. The authors are encouraged to show the effect of sgHUWE1 on TRIM52 turnover in a CHX chase experiment. Additionally, the band intensity difference in the sgUBR4 blot is noticeable compared to sgKCMF1 and sgBIRC6. Clarification is needed on whether UBR4 knockout affects TRIM52 morphology or proliferation.

Reply: The revised manuscript now contains CHX chase data for TRIM52 with a complete set of E3 ligase knock-outs, including HUWE1 (new data: Extended data Fig.5c-d). In line with our other data, individual knock-out of all of the E3 ligases increased the TRIM52 protein half-life by 2.5 - 5-fold. In this data set sgUBR4 results were comparable to the other E3 knock-outs; based on our observations we do not think that loss of UBR4 affects cell fitness at the time of analysis.

8. It would be beneficial to examine the autoubiquitination effect of the R187A mutant in an *in vivo* model and assess the stability of TRIM52 with this mutation.

Reply: Previous experiments from our lab have shown that RING inactivation through mutation of one of the zinc-coordinating cysteine residues did not affect TRIM52 turn-over (Hacker *et al.* 2019). To further address whether auto-ubiquitination is required for TRIM52 degradation, the half-life of exogenously expressed TRIM52-R187A was determined by CHX chase (new data, included as Extended data Fig.6f). Consistent with the cysteine mutant, the R187A mutant still had a short half-life of 31.6 min., which was comparable to its exogenously expressed wild-type counterpart (36.2 min.).

Minor comments:

- In Figure 2i, displaying the entire band of 3xHA TRIM52 would be beneficial to confirm the absence of ubiquitination in the KtoR mutant following Etoposide treatment.*

Reply: The updated figure panel (Fig. 3i) shows a larger part of the blot. In our experience, it is difficult to see specific ubiquitinated species without prior enrichment through IP or pull-down. Although we cannot exclude that some of the smearing above the TRIM52 band is poly-ubiquitin, the fact that epoxomicin treatment did not increase smearing relative to the amount of TRIM52, makes it likely that this is rather non-specific signal. In fact, proper ubiquitination analysis of the KtoR mutant by IP (Fig. 3f) showed that despite substantially increased protein levels of the KtoR mutant, its ubiquitination was significantly decreased.

i

- There are a few spelling errors that need correction: In Figure 2f, "ubiquitin" is misspelled, in Figure 2h, "identified" is misspelled in the second bullet point, and in the Figure 4 legends (g), "form" should be corrected to "from."*

Reply: The mistakes have been rectified in the revised manuscript.

11. The abbreviations for Etoposide and Epoxomicin are unclear. It would be helpful to define these abbreviations in the main text or figure legends.

Reply: The abbreviations have been defined in the text upon first use.

12. For Figure 3g, please include larger images of the TRIM52 blot. Additionally, the font size in the labels on the right side needs to be increased, and the text "endogenous TRIM52" should be corrected.

Reply: The TRIM52 image was increased, and the font size of the labeling increased.

13. In Figure 5a, clarification is needed on the difference between lanes 4 and 5, as the dots indicate the same genes and enzymes.

Reply: The difference between lanes 4 and 5 is the use of different E1s for the reaction; Supplementary data 6 includes the full list of proteins used in *in vitro* ubiquitination reactions from Fig.6 and Extended data Fig.7.

14. Finally, in Figure 5f, there is a discrepancy in the GFP blot compared to Figure 5g. A higher exposure image of the blot would be appreciated to provide more precise results.

Reply: The updated figure panel contains a higher exposure version of the EGFP blot

Reviewer #4:

Major points

1. *The DEPMAP database shows that TRIM52 is non-essential in RKO cells. Though I am happy to concede that DEPMAP may be wrong, the authors need to rescue their Δ TRIM52 cell proliferation defect by reintroducing exogenous TRIM52.*

Reply: We had indeed noticed ourselves as well that *TRIM52* does not have a negative score in RKO cells in the DEPMAP CRISPR panel, while it does so moderately in the RNAi panel. We think that this could stem from *TRIM52*-depletion efficiency, as we have observed for both CRISPR- and RNAi-mediated loss of *TRIM52*, that only sgRNAs and shRNAs that nearly completely depleted the *TRIM52* protein had strong loss-of-fitness phenotypes.

Prior to the initial submission and during the revision process we extensively invested in various approaches to functionally rescue *TRIM52* knock-out (sgRNA) and knock-down (shRNA) cells with wild-type and mutant sgRNA/shRNA-resistant *TRIM52* cDNA expression constructs. We attempted expression of tagged and untagged *TRIM52* versions, different promoter strengths for driving transgene expression of all reported *TRIM52* isoforms. Unfortunately, none of these tested conditions functionally rescued fitness of *TRIM52* loss-of-function cells in competition experiments.

Although we do not have definitively determined why these rescue experiments did not work, we speculate that substantially higher transgene expression than endogenous *TRIM52* (more than 10-fold even with the weakest tested promoter) could hinder functional rescue.

We want to emphasize that our results show that our functional *TRIM52* data are specific (yet likely require endogenous *TRIM52* levels), whereas *TRIM52* protein degradation phenocopies the endogenous protein independent of expression levels and tags. Specifically, we have targeted *TRIM52* with three independent shRNAs (Benke *et al.* Oncotarget 2018), and seven independent sgRNAs (Hacker *et al.* Scientific Reports 2019, and current manuscript), which all reduced cell fitness. Moreover, the magnitude of fitness-loss scaled with the strength of *TRIM52* protein depletion. This strongly indicates that the measured loss-of-*TRIM52* effects are specific and not the results of off-target effects.

In summary, it could be that the CRISPR conditions curated in DEPMAP for RKO cells do not represent strong depletion of *TRIM52*, and hence did not result in a strong loss-of-fitness phenotype. We intend to address this in future experiments by driving rescue construct expression from the endogenous *TRIM52* promoter/locus.

2. *From their synthetic lethal CRISPR screen, the authors find many DDR genes that synergize with the loss of TRIM52. What's perplexing is that these DDR genes aren't really interrelated but a smattering of DDR factors that work to repair distinct types of DNA damage (e.g. replication stress-induced breaks, mismatches, double-stranded breaks). Additionally, while NHEJ1 is their top hit, very few of the other canonical members of the NHEJ pathway, outside of XRCC4, came up as hits (e.g. KU70, KU80, etc). While it's*

possible that these genes, being essential, might have dropped out of the screen, the authors should perform competition assays to validate whether the loss of other NHEJ genes also compromises cell fitness in the context of *TRIM52* loss. Further, there is very little overlap between this gene set and their unbiased proximity-labeled screen (Fig. 1h), complicating the biological relevance of their hits.

Reply: Additional competition assays were performed with cells in which individual HDR (RAD51 and BARD1) or NHEJ (Ku70 and Ku80) components were targeted, in combination with *TRIM52* ablation. These new data were added as Extended Data Figure 1a-b to the revised manuscript. In contrast to our previous speculation that *TRIM52* could function upstream of the HDR pathway, loss of both NHEJ and HDR components had additive loss-of-fitness effects to *TRIM52* ablation. These non-epistatic interactions with both the NHEJ and HDR pathways suggest that *TRIM52* can redundantly function in parallel to either repair pathway, in a partially functionally redundant manner. Both HDR and NHEJ have been reported to repair TOP2-induced lesions, based on these findings we therefore hypothesize that *TRIM52* acts in parallel to both of these pathways, and that in the absence of *TRIM52* cells rely significantly more on these DNA repair pathways to resolve TOP2-mediated damage. The manuscript text has been adjusted to reflect our new findings and interpretation of the data.

The fact that core components of neither repair pathway were identified as hits in the suppressor screen is consistent with the pre-dropout screening strategy, which removed cells targeting essential genes.

3. The biggest flaw with this manuscript is that the authors use a defect in cell proliferation as a proxy for DNA damage (Fig. 1g). Though the authors show that loss of *TRIM52* synergizes with *DDR* genes, this synergy could have nothing to do with DNA damage. The authors need to show that loss of *TRIM52* results in DNA damage (e.g. alkaline comet assays, γ H2AX staining). What's more puzzling is that the author's previous publication (Benke et al. 2018) shows that loss of *TRIM52* causes cells to exit from the cell cycle and remain in a G0/G1 state. Thus, their observed proliferation defect could simply be due to this cell cycle exit. This huge unknown makes evaluating the manuscript premature. For example, if loss of *TRIM52* causes DNA damage, what types of breaks does it make? Which substrates of the *DDR* signaling pathway are regulated by *TRIM52*? Is the ligase activity of *TRIM52* required during the *DDR* or does *TRIM52* have ligase-independent functions? If loss of *TRIM52* does not cause DNA damage, on the other hand, what is the biological relevance of the observed synergy between the *TRIM52* and *DDR* genes?

Reply: Additional experiments were performed to further analyze the role of *TRIM52* in restricting DNA damage. Data from these experiments were added as new Fig.2 f-g. To this end, sgAAVS1 or sg*TRIM52* cells treated with or without the TOP2 poison Etoposide were analyzed by γ H2AX staining. Upon ablation of *TRIM52*, we measured a trend (albeit non-significant) for an increased number of γ H2AX foci in DMSO-treated *TRIM52* KO cells. Importantly, in the presence of Etoposide, loss of *TRIM52* significantly increased the number of γ H2AX-marked DNA lesions. This result is consistent with our RADAR assay data that loss of *TRIM52* leads to a defect in the resolution of covalently associated TOP2 from the DNA, resulting in the formation of double strand breaks. Combined with our data presented in our manuscripts, we therefore propose that in the absence of *TRIM52*, TOP2-DNA damage activates p53, which in turn decreases proliferation.

4. The authors should show their studies using overexpressed TRIM52 (TurboID, puncta formation in the nucleus, etc) is not an overexpression artifact. Though they sought to match the expression of TRIM52 to GFP, they should have matched it to endogenous TRIM52. Or at least validate their proximity labeling studies in some way.

Reply: As explained in more detail above, we have been technically unable to exogenously express TRIM52 constructs at endogenous levels. Moreover, analysis of endogenous TRIM52 is technically challenging because of the extremely low steady-state levels, and the fact that the only available antibody only works in Western blot and not for IP or immuno-fluorescence microscopy.

In that context, the revised manuscript contains two new data sets to address this comment. Firstly, subcellular fractionation was performed and showed that the endogenous TRIM52 accumulates in the nucleus in response to epoxomicin treatment (Extended Data Fig. 2g), consistent with what we have shown for the TurboID-TRIM52 construct. Secondly, we compared by immune-fluorescence assay, localization of TurboID-TRIM52 with EGFP-TRIM52 and mCherry-TRIM52, and found that all localize to the nucleus upon inhibition of proteasomal degradation with epoxomicin.

5. The authors nicely show, by using their cellular fluorescent reporter system, that loss of HUWE1, BIRC6, and UBR4 causes stabilization of the GFP-TRIM52 reporter. Yet, their *in vitro* reconstitution of TRIM52 ubiquitination remains unconvincing (Figs. 5e and f). Figures 5e and f show that HUWE1, BIRC6, and UBR4 are active ubiquitin ligases and BIRC6 likely monoubiquitates TRIM52. However, none of the ligases seem to polyubiquitate TRIM52 in their purified system, given there is a complete lack of higher-order molecular weight species or a concomitant loss of the unmodified form in their aGFP blots. The ubiquitin chains do not seem to be covalently linked to TRIM52 but co-IP indirectly.

Reply: A TRIM52 detection panel was added to Fig. 6e in the revised manuscript. In this experiment, EGFP-TRIM52 was mono-ubiquitinated with BIRC6, and subsequently introduced as a substrate in a reaction with KCMF1/UBR4. Under these conditions, UBR4/KCMF1 generate higher-MW species on EGFP-TRIM52, both in the anti-EGFP and anti-TRIM52 blots. Together, these results support the hypothesis that monoubiquitination seeded on TRIM52 by BIRC6 is extended into covalently attached polyubiquitin chains by UBR4/KCMF1.

Minor points

6. *Why are there no polyubiquitin chains in the mCherry IP for the MYC-mCherry-Trim52 construct for Figure 2g when these chains are clearly visible in Fig. 5e?*

Reply: In former Fig. 2g (now Fig. 3g) there are polyubiquitin chains in the MYC-mCherry-TRIM52 sample co-transfected with His-ubiquitin. We could not detect polyubiquitin chains in the control sample without His-Ubiquitin, because the signal from endogenous ubiquitination is much weaker and cannot be detected on the same blot as samples with overexpressed His-ubiquitin. The ubiquitination of TRIM52 by endogenous ubiquitin can be seen in Fig. 3e-f. The chains visible in Fig. 6e are ubiquitin chains from an *in vitro* ubiquitination reaction with purified TRIM52, E3 ligases and FLAG-labelled ubiquitin.

7. *Which lysines were mutated in the 5KtoR construct? Did the authors identify the incorrect lysines (Figure 2l)?*

Reply: In the 5KtoR construct we mutated the two lysines identified to be ubiquitinated by nLC-MS/MS (K50, K256), as well as three lysines that were not covered by peptides in the nLC-MS/MS analysis (K28, K288, K235). The mutation of the identified residues and the additional three lysines did not result in TRIM52 stabilization. Only mutation of all lysines in the protein increased steady-state TRIM52 levels. These results suggest that the E3 ligase machinery targeting TRIM52 could be able to modify non-dominant lysines in their absence.

Rebuttal letter for Nature Communications manuscript NCOMMS-24-30996-A

" TRIM52 maintains genomic DNA integrity and is under tight proteolytic control by multiple giant E3 ligases"
Alexandra Shulkina *et al.*

We would like to very much thank the reviewers for their efforts and constructive feedback on our revised manuscript. Please find below a point-by-point reply to the two remaining reviewer comments.

Point-by-point rebuttal

Reviewer comments are in *blue cursive*.

We confirm that the reviewer comments are in full and reproduced verbatim as provided by the editorial office.

REVIEWER COMMENTS

Reviewer #1 (Remarks to the Author):

The authors have satisfactorily addressed all concerns raised. The manuscript is now suitable for publication.

Reply: We thank this reviewer very much for their time and effort to review our manuscript and provide constructive feedback.

Reviewer #2 (Remarks to the Author):

Reply: We are grateful to this co-reviewer for their useful feedback on our work.

Reviewer #3 (Remarks to the Author):

The authors have significantly improved the revised version with the additional set of experiments and clarified some aspects of their findings in the text. They have also addressed my concerns on their first version. I am happy to endorse this revised manuscript for publication in Nat Comm.

Reply:

(This is the first reply to the second feedback on the revised manuscript from reviewer #3)

We thank this reviewer very much for their time and effort to review our manuscript and provide constructive feedback.

Reviewer #3 (Remarks to the Author)

The authors have significantly improved the revised version with the additional set of experiments, clarified some aspects of their findings in the text, and addressed some of my concerns about their first version. However, the revised version fails to convince me that TRIM52 is involved in Top2-cc repair. If the authors want to emphasize that TRIM52 indeed works as one of the pathways for Top2cc repair, they have to provide more solid data. In addition, Fig 2f is of very poor resolution, and nuclei are much larger in sgTRIM52/ETO treated cells, than in sgAAVS1/ETO treated cells; so the number of increased γ -H2AX foci/nucleus could be significant as the nuclei are just simply larger in sgTRIM52/ETO than in sgAAVS1/ETO.

Reply:

(This is the reply to the second feedback on the revised manuscript from reviewer #3)

The limited resolution of Fig. 2F may have come from the conversion to PDF. To further address this point, also some of the IF panels were replaced by images with better resolution.

The reviewer is correct that indeed the nuclei of *TRIM52*-targeted cells are larger. This is quantified in the graph below (which is not included in the revised manuscript).

To address whether the increased number of γ H2AX foci measured stemmed from the larger nuclei in *TRIM52* KO cells, we normalized the γ H2AX focus numbers to their respective nuclear surface areas. This normalized quantification was added to the revised manuscript as Figure 2H. These data show that in line with our RADAR and absolute γ H2AX foci quantifications, loss of *TRIM52* significantly increased the number of γ H2AX foci normalized by nuclear surface area. From this analysis we concluded that the increased nuclear surface area is unlikely to explain the increased γ H2AX signal upon *TRIM52* loss. This is consistent with increased TOP2cc detection by RADAR assay, and increased levels of activated phospho-p53 and cell cycle arrest in *TRIM52*-targeted cells (Benke *et al.* 2018).

Regardless, the author should perform a better set of experiments to convince me that *TRIM52* is indeed involved in Top2-cc repair.

Reply: Our data in Fig. 2 show that upon *TRIM52* loss, TOP2cc and γ H2AX are increased. Our interpretation of the experiments suggested by this reviewer below is that they would address more mechanistically which specific step of limiting TOP2cc complexes is controlled by *TRIM52*. In particular whether *TRIM52* plays a direct mechanistic role in resolution of TOP2cc complexes, and how it does so.

We recognize that understanding how *TRIM52* mechanistically is required to maintain genome integrity and prevent ultimate cell cycle arrest is of great interest. Although we aim to investigate this in future studies, it is our opinion that it falls outside the scope of the current manuscript. Nevertheless, these experiments are in our opinion excellent suggestions, which we aim to pursue in the future.

At several instances in the manuscript we formulated conclusions from our data as *TRIM52* having a role in ‘the resolution of TOP2cc complexes’. We aimed to convey that *TRIM52* likely plays a role somewhere between the formation of TOP2cc complexes and their ultimate resolution. However, we recognize that this may have given the unsupported impression that *TRIM52* plays a direct mechanistic role in a resolution step. To remove any

impression of such a mechanistic claim, we rephrased any text in the manuscript that may have been interpreted as a claim that TRIM52 plays a direct mechanistic role in resolving TOP2 lesions.

In particular we rephrased the following text parts:

Abstract: lines 24-25

Introduction: line 47
line 63
line 66

Results: lines 121-122
line 139
line 171
line 381

Discussion lines 475-476
line 478
line 497
line 504
lines 555-556

For instance:

1) The authors must perform RADAR assay with ETO treatment (to induce a high level of TOP2cc) and then remove the drug and monitor TOP2cc repair during the recovery time (e.g. 1 hr, 2 hr, 4 hr and 8 hr). This would be the 1st body of evidence that TRIM52 plays a role in TOP2cc resolution.

Reply: This would be an excellent experiment to address the mechanistic role of TRIM52 in the resolution phase, which in our opinion falls outside the scope for the current manuscript. We are very grateful for the thoughtful suggestion, which we aim to address in future work.

2) The cell cycle analysis (FACS) must support these experiments to show the effect of TRIM25-ko on the cell cycle under unperturbed conditions, in response to ETO and during the recovery time.

Reply: We have previously performed and reported part of this proposed flow cytometry experiment in the absence of ETO treatment (Benke *et al.* 2018). This experiment showed that in the absence of *TRIM52*, cells activate phospho-p53 and arrest in G0. Adding a recovery phase would be an excellent experiment to address the importance of TRIM52 in the resolution phase, which in our opinion falls outside the scope for the current manuscript. We are very grateful for the thoughtful suggestion, which we aim to address in future work.

3) These FACS data should be accompanied by a γ -H2AX signal to demonstrate the level of DSB activation. Alternatively, the authors can perform a Comet assay to show that DSBs persist or their repair is delayed in TRIM25-ko cells (during the recovery time), and these breaks depend on TOP2cc (after ETO treatment).

Reply: This would be an excellent experiment to address the mechanistic role of TRIM52 in the resolution phase, which in our opinion falls outside the scope for the current manuscript. We are very grateful for the thoughtful suggestion, which we aim to address in future work.

4) They also have to perform a CFA assay and confirm that TRIM52 cells are hypersensitive to ETO, and this hypersensitivity is a) abolished when TOP2 is removed (e.g. si or shTOP2) and b) restored when TRIM52wt is expressed back.

Reply: We aim to address hyper-sensitivity of *TRIM52* KO cells to ETO in future studies. CFA will be tried, but may turn out to not be technically feasible as practically all cells with a good loss of *TRIM52* will all be out-selected in ten days (Fig. 1C). The remaining ~10-15% of cells at day 10 (Fig. 1C) are likely cells that do not have a functional *TRIM52* knock-out. Given the strong, almost complete loss of cell fitness upon *TRIM52* knock-out, it is therefore uncertain whether additional sensitivity could be measured in CFAs given the long time needed for colonies to grow out. Instead we would favor similar sensitization assays in the presence of sublethal ETO concentrations with faster read-outs, such as cell cycle phase distribution by FACS (as indicated above) and cell competition assays as shown in Fig. 1C.

5) If TOP2cc indeed persists in *TRIM52* cells, they should also be able to visualize the persistence of DSBs by IF showing a γ -H2AX/53BP1 colocalization signal.

Reply: We are very grateful for the thoughtful suggestion, which we aim to address in future work.

6) Throughout these experiments, a positive control for abolished TOP2cc repair (e.g., siZNF45 or Rad54L2) must also be included.

Reply: We are very grateful for the thoughtful suggestion, we aim to include these controls in future work.

7) I suggest they check this recent paper on Top2cc repair: <https://www.science.org/doi/10.1126/sciadv.adl2108> and use it as an example how to demonstrate/confirm the potential role of *TRIM52* in Top2cc repair.

Reply: We are very grateful for the thoughtful reference, which we will use to carefully plan our future experiments aimed at understanding how *TRIM52* is mechanistically important for maintaining cell fitness in the presence and absence of TOP2 poisons.

Reviewer #4 (Remarks to the Author):

Shulkina and colleagues seek to decipher the functional relevance of TRIM52 in DNA repair biology and better understand the regulation underlying TRIM52 stability. The authors claim that TRIM52 is involved in DNA repair because TRIM52 “synergizes” with multiple DNA repair factors in their CRISPR screen. However, the biological role of TRIM52 and its mechanism of action in DNA repair remain unresolved.

1. Generally speaking, the loss of essential DNA repair factors causes cell death by inducing genomic instability in the form of irreversible DNA breaks, and the upregulation of gamma-H2AX is often used as an indicator of DNA damage and cell death if the damage is not resolved in a timely manner. The authors show that TRIM52 is essential (Fig. 1g), yet its loss does not increase DNA damage (Figs. 1d-g) without the addition of an external source of DNA damage. This indicates to me that the essentiality of TRIM52 is not due to its role in DNA repair but through another function (perhaps DNA replication). This issue makes their claims about epistatic relationships in the context of DNA repair unclear (lines 104-110).

Reply: We acknowledge that in the absence of etoposide we did not measure a significant difference in these DNA damage read-outs. However, in all measurements there is a clear trend of an increase in RADAR and γ H2AX signals upon *TRIM52* loss in the absence of etoposide (Fig. 2d-g). Although we cannot be certain given the lack of statistical significance in the non-stimulated samples, these observations are consistent with a relative slow activation of p53 and cell-cycle arrest in *TRIM52* knock-out cells as a consequence of slow DNA damage accumulation during cellular genome replication and/or transcription (this manuscript, and Benke *et al.* 2018). Importantly, *TRIM52* knock-out or knock-down cells are not lost as a consequence of cell death, but exclusively by cell cycle arrest (Benke *et al.* 2018).

Our interpretation of the data in non-stimulated samples (Fig 2d-g) is that while there is a trend towards an increase in DNA damage markers upon *TRIM52* knock-out, likely a combination of the effect size, experimental variance, and/or number of replicates has limited the statistical power. Importantly, in all instances in which we measured the effect of *TRIM52* loss by different DNA damage read-outs (RADAR, γ H2AX)

we see statistically significant increases in these DNA damage markers specifically in the presence of etoposide, indicating a TOP2-related effect of *TRIM52* loss.

We have added text in the discussion (lines 473-484) to underpin this caveat regarding the non-stimulated samples in the context of the double-knock-out experiments.

Added text: “It should be noted that while loss of *TRIM52* in the absence of etoposide showed a consistent and reproducible trend towards increased DNA damage markers (Fig. 2d-g), these changes were non-significant. We speculate that a combination of the effect size, experimental variance, and/or number of replicates may underlie the limited statistical power in these samples. Nevertheless, low increases in DNA damage from endogenous sources such as genome replication or transcription are consistent with relatively slow p53 activation and cell cycle arrest upon *TRIM52* loss⁸. In the *TRIM52* double-knock-out experiments with key components of the NHEJ and HDR branches (Extended Data Fig. 1a-b) the effect on cell fitness was measured. From these experiments we concluded that both NHEJ and HDR are functionally redundant for *TRIM52* cellular output. These results are consistent with cell-fitness loss stemming from DNA damage in the *TRIM52* knock-out cells. However, further future examination of DNA damage markers in non-stimulated *TRIM52* knock-out cells will be important to further substantiate this conclusion.”

2. I remain unconvinced by the author's in vitro reconstitution experiments (Fig. 6F). The authors claim that BIRC6 and UBR4 work together to polyubiquitinate GFP-TRIM52, yet I do not see any higher molecular weight species in their anti-GFP blots (either in their inputs or IPs). Curiously, the authors instead blot for ubiquitin (anti-FLAG) and find an increase in ubiquitin signal only when they IP their substrate under native IP conditions. This simply tells me that ubiquitin chains co-associate with GFP-TRIM52 indirectly instead of being covalently linked to GFP-TRIM52 (because there are no higher molecular weight species in the corresponding anti-GFP blots). If the authors want to show that BIRC6/HUWE1 are covalently modifying GFP-TRIM52 with ubiquitin chains using this type of assay, they really need to perform their IPs under denaturing (or non-native) conditions.

Reply: We acknowledge that with the available exposures of the GFP panel in Fig. 6f we could not detect higher molecular weight species of *TRIM52*. We think this stems from only a smaller fraction of the total *TRIM52* pool being poly-ubiquitinated by the UBR4 complex. However, the right most samples in Fig. 6e show increased higher molecular weight species in the *TRIM52* and GFP panels, predominantly above the size of *TRIM52*, consistent with covalent ubiquitination of *TRIM52* by UBR4.

However, in line with this reviewer's assessment, we are also of the opinion that there is some smaller part of the ubiquitin signal originating for non-covalently bound free ubiquitin. Our conclusion of these results is that there likely is a mix of some free unbound, and specifically covalent *TRIM52* bound ubiquitin. Consistent with this conclusion, there is limited signal from lower MW signal below *TRIM52*, yet stronger ubiquitin signal above the *TRIM52* MW.

We have added extra text to the results section (lines 440-446) to point out this caveat and add nuance to our interpretation of the data.

“A limited amount of ubiquitin signal below the MW of *TRIM52* suggests that the samples contained a small amount of non-covalently bound free ubiquitin chains (Fig. 6e-f). However, the stronger ubiquitin signal above the *TRIM52* MW suggested that a substantial fraction of the poly-ubiquitin chains was covalently attached to *TRIM52*. It should be noted that the limited amount of high molecular weight species in the GFP/*TRIM52* blots indicates that under these reaction conditions, UBR4-dependent poly-ubiquitination of mono-ubiquitinated *TRIM52* has limited efficiency *in vitro*.”

Reviewer #5 (Remarks to the Author):

Reply: We are grateful to this co-reviewer for their time and constructive feedback on our work.